# QUAMBA: A POST-TRAINING QUANTIZATION RECIPE FOR SELECTIVE STATE SPACE MODELS

**Hung-Yueh Chiang**[*1] , **Chi-Chih Chang**[*†2,3] , **Natalia Frumkin**[1] ,
**Kai-Chiang Wu**[2] , **Diana Marculescu**[1]

[1] Chandra Family Department of Electrical and Computer Engineering,
  The University of Texas at Austin
[2] Department of Computer Science, National Yang Ming Chiao Tung University
[3] Department of Electrical and Computer Engineering, Cornell University
`{hungyueh.chiang, nfrumkin, dianam}@utexas.edu,`
`cc2869@cornell.edu, kcw@cs.nycu.edu.tw`

## ABSTRACT

State Space Models (SSMs) have emerged as an appealing alternative to Transformers for large language models, achieving state-of-the-art accuracy with constant memory complexity which allows for holding longer context lengths than attention-based networks. The superior computational efficiency of SSMs in long sequence modeling positions them favorably over Transformers in many scenarios. However, improving the efficiency of SSMs on request-intensive cloud-serving and resource-limited edge applications is still a formidable task. SSM quantization is a possible solution to this problem, making SSMs more suitable for wide deployment, while still maintaining their accuracy. Quantization is a common technique to reduce the model size and to utilize the low bit-width acceleration features on modern computing units, yet existing quantization techniques are poorly suited for SSMs. Most notably, SSMs have highly sensitive feature maps within the selective scan mechanism (*i.e.,* linear recurrence) and *massive outliers* in the output activations which are not present in the output of token-mixing in the self-attention modules. To address this issue, we propose a *static* 8-bit per-tensor SSM quantization method which suppresses the maximum values of the input activations to the selective SSM for finer quantization precision and quantizes the output activations in an outlier-free space with Hadamard transform. Our 8-bit weight-activation quantized Mamba 2.8B SSM benefits from hardware acceleration and achieves a $1.72 \times$ lower generation latency on an Nvidia Orin Nano 8G, with only a 0.9% drop in average accuracy on zero-shot tasks. When quantizing Jamba, a 52B parameter SSM-style language model, we observe only a $1\%$ drop in accuracy, demonstrating that our SSM quantization method is both effective and scalable for large language models, which require appropriate compression techniques for deployment. The experiments demonstrate the effectiveness and practical applicability of our approach for deploying SSM-based models of all sizes on both cloud and edge platforms. Code is released at `https://github.com/enyac-group/Quamba`.

## 1 INTRODUCTION

State Space Models (SSMs) (Gu & Dao, 2023; Lieber et al., 2024b) have attracted notable attention due to their efficiency in long sequence modeling and comparable performance to Transformers (Zhang et al., 2022; Brown et al., 2020). Although Transformers have shown a strong ability to capture the causal relationships in long sequences, the self-attention module within Transformers incurs a quadratic computation complexity with respect to the context length in the prefilling stage as well as a linear memory complexity (*e.g.,* the K-V cache) in the generation stage. In contrast,

---

[*]Equal contribution.
[†]The work was done at National Yang Ming Chiao Tung University

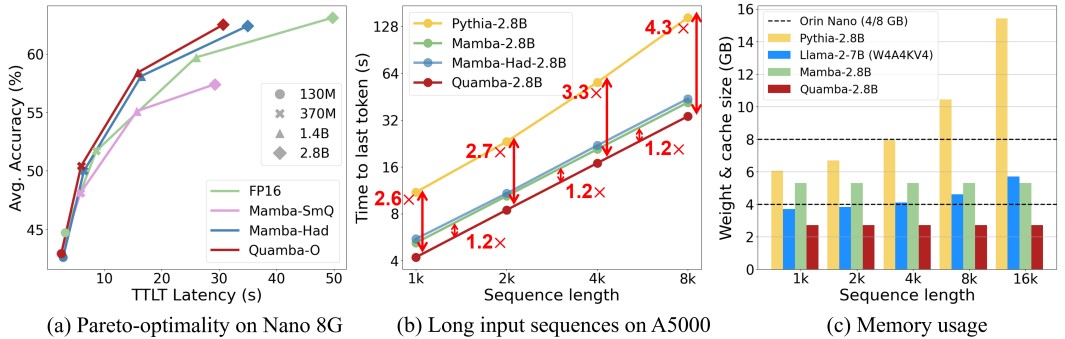

(a) Pareto-optimality on Nano 8G     (b) Long input sequences on A5000     (c) Memory usage

Figure 1: We demonstrate that (a) our method achieves Pareto-optimality on the Nano 8G with 1K input tokens. Figure (b) shows latency speedups for long input sequences on the A5000, and Figure (c) shows the memory usage across devices comparing to Pythia 2.8B (Biderman et al., 2023) and 4-bit Llama-2-7B (Touvron et al., 2023).

SSMs, an attractive substitute to Transformers, perform sequence modeling with a recurrent neural network-like (RNN-like) linear recurrence module (*i.e.,* selective scan) which has *linear* computation complexity in the prefilling stage and *constant* memory complexity in the generation stage.

Despite their computational advantages, deploying SSMs on diverse hardware is challenging due to memory and latency constraints. Quantization, such as 8-bit integer (INT8), offers a solution by reducing model size and latency while preserving accuracy and enabling hardware acceleration. Quantizing SSMs is non-trivial as current post-training quantization (PTQ) techniques for Transformers (Xiao et al., 2023; Dettmers et al., 2022) fail to handle the sensitive activations of the selective scan (*i.e.*, linear recurrence) resulting in poor performance (Lieber et al., 2024b; Zhao et al., 2024). Our study shows SSMs exhibit *distinct* outlier patterns compared to Transformers (Ashkboos et al., 2024b; Dettmers et al., 2022; Xiao et al., 2023; Zhao et al., 2023). We show that outliers appear in the SSMs output (*i.e.* the $y$ tensor). In contrast, the input and output of self-attention layers are relatively *smooth* and *do not* exhibit outlier issues. More comparisons can be found in Section M. This underscores the need for specialized quantization methods for SSM-based models, as their lack leads to suboptimal memory usage and latency on deployed hardware.

We first analyze the input and output activations of the selective scan module to reveal the quantization sensitivity and outliers in SSM activations (*ref.* Figure 2 and Figure 3). Specifically, we show that quantizing SSMs is particularly challenging since SSM input and output activations present a causal relationship, making the input tensor (*i.e.,* $x$ in Eq. 1) sensitive to quantization errors. Additionally, representing the large outliers using 8-bit precision in the output activations (*i.e.,* $y$ in Eq. 1), which

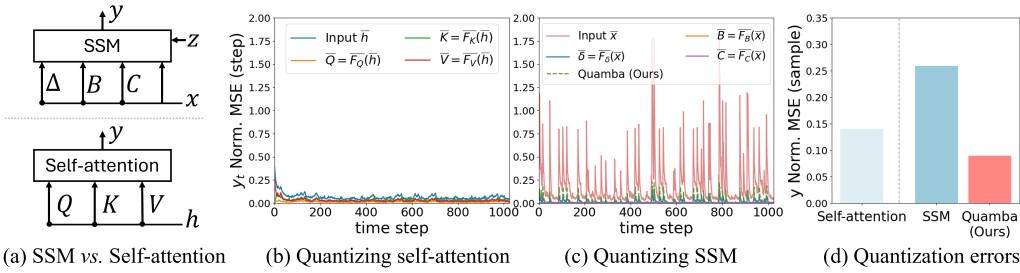

(a) SSM *vs.* Self-attention     (b) Quantizing self-attention     (c) Quantizing SSM     (d) Quantization errors

Figure 2: We analyze the sensitivity of quantization errors for (b) self-attention layers and (c) SSMs input activations. Our study shows that the $x$ tensor causes huge errors at the output $y$ due to the causal relationship of the linear recurrence, which is unique to SSMs. Self-attention layers are more robust to quantization errors. Our method (d) reduces the quantization error for the input sample. In Figure 12, we highlight the smooth, outlier, and sensitive paths in SSMs and self-attention layers.

are *not* present in the token-mixing output of self-attention modules, is difficult. To address this, we propose Quamba, an 8-bit *static per-tensor* quantization method for selective SSMs that can leverage the low bit-width acceleration features on modern computing units with *minimal overhead*. We suppress maximum values in input activations to SSMs, which is the most sensitive to the quantization error, for finer quantization precision. For the extreme outliers in output activations from SSMs, we use the Hadamard transform to smooth out the activations. Our quantized 8-bit 2.8B Mamba SSM achieves $1.72\times$ speedup in generation latency (*i.e.,* time-per-output-token, TPOT) on Nvidia Orin Nano 8G while only incurring a 0.9% accuracy drop in zero-shot tasks. We demonstrate our method achieves Pareto-optimality on the Nano 8G and lower the latency by $1.2 \times$ on A5000, as shown in Figure 1. While quantizing Jamba, a 52B parameter SSM-style language model, we observe just a $\sim 1\%$ reduction in accuracy. The effectiveness and scalability of our quantization technique for SSM-based models, as well as the practicality of our approach for deploying SSM-based models of various sizes on cloud and edge platforms.

## 2 RELATED WORK

**Model quantization.** Deploying large, over-parameterized neural networks on resource-constrained devices is challenging, and researchers have developed model quantization (Han et al., 2015; Jacob et al., 2018; Wang et al., 2019) as a solution to this problem. Quantization techniques reduce the data type precision (*e.g.* FP32 to INT4) to compress the model size and accelerate inference. The quantization techniques are generally divided into two categories: Quantization-aware training (QAT) and post-training quantization (PTQ) (Zhu et al., 2023; Gholami et al., 2022; Zhou et al., 2024). QAT (Liu et al., 2023; Dettmers et al., 2024) requires additional training efforts to adapt models to low bit-width, in exchange for better model performance. Our work falls under PTQ, which does not require training and can therefore be plug-and-play.

**LLM post-training quantization.** Post-training quantization (PTQ) techniques are generally broken down into two categories: weight-only quantization (*e.g.,* W4A16 and W2A16) and weight-activation quantization (*e.g.,* W8A8 and W4A4) (Zhu et al., 2023). Weight-only quantization (Frantar et al., 2022; Lin et al., 2023) focuses on quantizing weight matrices (*e.g.,* 4-bit or 2-bit) while keeping the activations in half-precision floating point. However, although weight-only quantization reduces memory usage, it still requires costly floating-point arithmetic for compute-intensive operations (*e.g.,* linear layers). To utilize low bit-width operations, Xiao et al. (2023); Zhao et al. (2023); Ashkboos et al. (2024b); Dettmers et al. (2022) study quantization for both weights and activations in Transformers. They address outliers in activations by using mixed-precision (Dettmers et al., 2022), rescaling quantization factors (Xiao et al., 2023), group quantization (Zhao et al., 2023), and quantizing activations in an outlier-free space (Ashkboos et al., 2024b). Unfortunately, these techniques target Transformers, do not generalize well to SSMs, and either fail to handle the sensitive tensors in SSMs resulting in poor performance (Xiao et al., 2023; Dettmers et al., 2022) or introduce additional computational overhead to the input of the selective scan (Zhao et al., 2023; Ashkboos et al., 2024b). Our research addresses this gap by examining SSM weight-activation quantization, aiming to concurrently reduce memory and compute costs by harnessing hardware acceleration for integer operations.

**State Space Models.** In recent times, a new wave of RNN-like models (Gu et al., 2021; Smith et al., 2022; Peng et al., 2023; Gu & Dao, 2023; Beck et al., 2024) has emerged, noted for their efficacy in modeling long-range dependencies and achieving performance comparable to Transformers. State Space Models (SSMs) (Gu et al., 2021; Smith et al., 2022; Gu & Dao, 2023) are a promising class of architectures that have been successfully applied to various applications, such as text (Gu & Dao, 2023; Wang et al., 2024), image (Zhu et al., 2024; Nguyen et al., 2022), video (Li et al., 2024; Nguyen et al., 2022), and audio (Goel et al., 2022; Saon et al., 2023). Despite their successes, the challenge of deploying SSMs across resource-limited hardware platforms remains largely unresolved. Our work addresses this challenge by proposing a quantization method specifically tailored for SSMs.

## 3 BACKGROUND

### 3.1 SELECTIVE STATE SPACE MODELS

**State Space Models.** Inspired by continuous systems, discrete linear time-invariant (LTI) SSMs (Gu et al., 2020; 2021; 2022) map input sequences $\mathbf{x}$ to output sequences $\mathbf{y}$. Given a discrete input

signal $x_t$ at time step $t$, the transformation $x_t \mapsto y_t$ through a hidden state $h$ is defined as

$$h_t = \dot{A}h_{t-1} + \dot{B}x_t, \quad y_t = Ch_t + Dx_t \tag{1}$$

where $\dot{A}$ and $\dot{B}$ are discrete parameters. The discretization function for $\dot{A}$ and $\dot{B}$ with a given $\Delta$ is defined as $\dot{A} = \exp(\Delta A)$, $\dot{B} = (\Delta A)^{-1}(\exp(\Delta A) - I) \cdot \Delta B \approx \Delta B$. This system uses $A$ as a state transition parameter and $B$ and $C$ as projection parameters. $\Delta$ is the time-scale parameter that is used to discretize the continuous parameters $A$ and $B$. $D$ is an optional residual parameter. $(A, B, C, D, \Delta)$ are trainable parameters. A residual branch $z_t$ is applied to the SSM output such that $y_t \cdot \text{SiLU}(z_t)$ before the output projection.

**SSMs with selection.** Gu & Dao (2023) improve SSMs with selection by letting their parameters $B$, $C$, and $\Delta$ be input-dependent, allowing the model to selectively remember or ignore inputs based on their content. Specifically, the interaction with input $x_t$ is defined as $B_t = \text{F}_B(x_t)$, $C_t = \text{F}_C(x_t)$, $\Delta_t = \text{softplus}(\text{F}_\Delta(x_t))$ where $\text{F}_B$ and $\text{F}_C$ are linear layers that map $x_t \mapsto B_t, C_t$. $\text{F}_\Delta$ use two consecutive projection layers, such that $\text{F}_\Delta = \text{Proj}(\text{Proj}(x)) + \text{bias}$. With the selection mechanism, the model has changed from time-invariant to time-varying.

## 3.2 QUANTIZATION

We focus on symmetric uniform quantization to approximate floating-point weights and activations with discrete 8-bit signed integers (*i.e.,* INT8) due to its hardware compatibility. The general symmetric uniform quantization function is defined as

$$\overline{X} = \text{clamp}\left(\left\lfloor \frac{X}{s} \right\rceil, -2^{N-1}, 2^{N-1} - 1\right), \quad s = \frac{\max(|X|)}{2^{N-1} - 1}, \tag{2}$$

where $\overline{X}$ represents the quantized weights or activations in INT8, $X$ is the input matrix in floating point, and $s$ is the scaling factor (*i.e.,* quantization step) that is determined by the target bit-width $N$ ($N = 8$ in our setting). The *static* scaling factor $s$ is pre-calibrated on a subset of data and is *fixed* during inference. We use the notation $X$ to represent the floating-point matrices, and $\overline{X}$ to represent their quantized matrices with their floating-point scaling factors $s_x$. For operators, we use $\overline{f}(\cdot)$ to represent the quantized version of the function $f(\cdot)$ (*i.e.,* the weights are quantized in the function $\overline{f}$).

## 3.3 WALSH–HADAMARD TRANSFORM

**Hadamard matrices.** A Hadamard matrix is an $n$-dimensional square matrix whose entries are either $+1$ or $-1$, and the rows and columns are mutually orthogonal with the computational property $\mathbf{H}_n\mathbf{H}_n^\top = n\mathbf{I}_n$. Walsh-Hadamard matrix is a special category of Hadamard matrix, consisting of square matrices of size $2^k$ and can be constructed as follows:

$$\mathbf{H}_{2^k} = \begin{bmatrix} \mathbf{H}_{2^{k-1}} & \mathbf{H}_{2^{k-1}} \\ \mathbf{H}_{2^{k-1}} & -\mathbf{H}_{2^{k-1}} \end{bmatrix} = \mathbf{H}_2 \otimes \mathbf{H}_{2^{k-1}} \quad , \text{where} \quad \mathbf{H}_2 = \begin{bmatrix} 1 & 1 \\ 1 & -1 \end{bmatrix}.$$

**Walsh–Hadamard transform.** The Walsh–Hadamard transform (WHT), a generalized class of Fourier transforms, has been applied to many related areas, such as LLM quantization (Ashkboos et al., 2024b) and efficient transfer learning (Yang et al., 2024), due to its favorable computational properties. We perform WHT to remove outliers from the output of the selective SSM. WHT projects a discrete input sequence signal onto a set of square waves (*i.e.,* Walsh functions). Its forward and inverse transform can be expressed in matrix form as $\tilde{x} = \mathbf{H}_n x$ , and $x = \mathbf{H}_n^\top \tilde{x}$. $x$ is the input discrete sequence signal, and $\tilde{x}$ denotes the WHT coefficients (*i.e.,* sequence components) that describe the magnitude of the square waves. WHT is efficient since the transform matrices consist only of real numbers, +1 or -1, so no multiplications are needed. The fast Walsh–Hadamard transform (FWHT) can compute the transformation with a complexity of $n\log n$ in a GPU-friendly (*i.e.,* parallelizable) fashion (Dao, 2024b). For input dimension $n \neq 2^k$, we factorize $n = 2^p m$, where $m$ is the size of a known Hadamard matrix (Sloane, 1999).

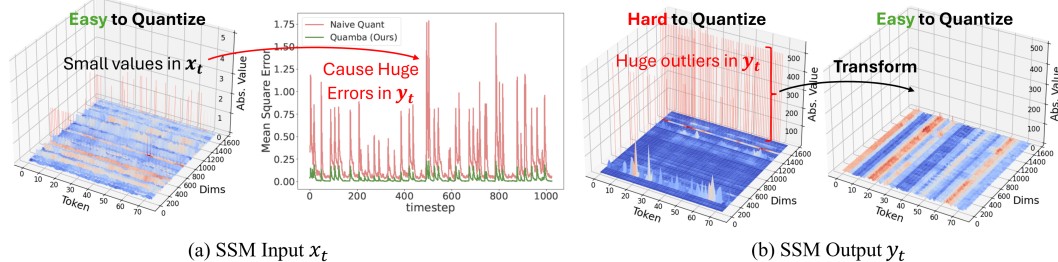

(a) SSM Input $x_t$ | (b) SSM Output $y_t$

Figure 3: The primary difficulties in quantizing Mamba blocks lie in the precision of the activations input into and output from the selective SSM. Although inputs are numerically small, the quantization step is skewed by the maximum value, causing significant errors in the output SSMs after the linear recurrent system. In contrast, large outliers are observed in the outputs. We use Hadamard matrices to transform the outputs to an outlier-free space.

# 4 QUAMBA: QUANTIZING MAMBA BLOCKS

## 4.1 PRELIMINARY STUDY

**Theoretical error bound for SSM quantization.** We consider a discrete linear time-invariant (LTI) state space model at time step $t$ defined as $h(t) = A(t)h(t-1) + Bx(t)$, where the state matrix $A(t) \in \mathbf{R}^{N \times N}$, input matrix $B \in \mathbf{R}^{N \times P}$, implicit latent state $h(t) \in \mathbf{R}^{N \times 1}$, and input $x(t) \in \mathbf{R}^{P \times 1}$. We assume the norm of the state matrix A can be bounded by an exponential function such that $||A(t)||_2 \le a \cdot e^{t-T}$, and the input matrix $B$ is bounded by $||B|| \le b$, where $0 < a < 1$, $b > 0$, and $1 \le t \le T$. In our proof, we utilize the spectral norm ($|| \cdot || = || \cdot ||_2$). Furthermore, we assume the system input contains a quantization error $\delta_x(t) \in \mathbf{R}^{P \times 1}$, such that $\overline{x}(t) = x(t) + \delta_x(t)$, where $||\delta_x(t)||_2 \le \epsilon$ and $\epsilon > 0$. The system is initialized as $h(0) = [0, 0, \dots 0]^T$. For details of the proof, please check Section N.2 in Appendix N.

**Theorem 4.1.** *The quantization error of $\Delta(t) = \overline{h}(t) - h(t)$ at time step $t$ for the given discrete linear time-invariant model is bounded such that: $||\Delta(t)||_2 \le \epsilon b \left( \frac{1}{1-ae^{t-T}} \right)$. Consequently, the* **global** *quantization error (i.e., $t = T$) is bounded by: $||\Delta(T)||_2 \le \frac{\epsilon b}{1-a}$.*

**Empirical analysis of SSM quantization.** As shown in Figure 2 and Figure 3, the main challenge in quantizing Mamba blocks is the precision of the SSM input activation $x$, which is sensitive to quantization errors, and the output activation $y$. In Figure 1 (a), the naive 8-bit quantization introduces large quantization errors, resulting in model collapse for all model sizes. We delve into the causal relationship between $(A, B, C, \Delta, x)$ and $y$ as modeled by the linear recurrent system in Equation 1. As shown in Figure 2 (b) and Figure 3 (a), we find that $x$ is sensitive to quantization errors and it leads to the largest errors in the SSM output $y$, although the values in the $x$ tensor are numerically small (ref: Figure 13). We conjecture the reason behind the phenomenon is that $(B, C, \Delta)$ are input-dependent (*i.e., $x$*-dependent). We note that this finding is specific to SSMs. As shown in Figure 2 (a), self-attention layers are more resilient to quantization errors and do not experience the same issues.

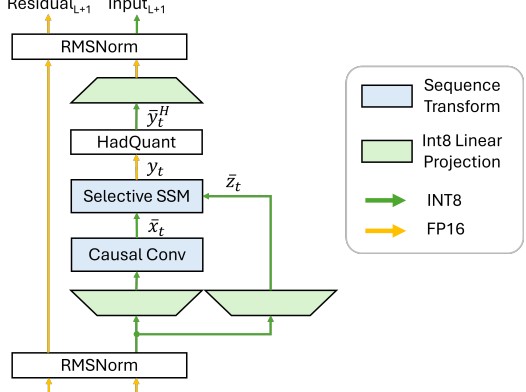

Figure 4: The precision mapping and dataflow of Quamba. All scaling factors and element-wise operations (*i.e.,* non-linearity and residual addition) are fused in the quantized operations.

**SSM outliers.** Our study shows SSMs exhibit distinct outlier patterns compared to Transformers (Ashkboos et al., 2024b; Dettmers et al., 2022; Xiao et al., 2023; Zhao et al., 2023). We show that outliers appear in the SSMs output (*i.e.* the $y$ tensor), which perform a similar token-mixing function to self-attention layers. In contrast, the input and output of self-attention layers are relatively smooth and do not exhibit outlier issues. More comparisons can be found in Section M. This highlights that different quantization methods are needed for SSM-based models. Therefore, we tailor two different quantization techniques: percentile-based quantization for the input activation $x$ and quantizing the output $y$ in an outlier-free space using the Hadamard transform. Our method recovers performance by improving the quantization precision for the inputs and outputs of the SSM and achieves a better trade-off between accuracy and latency.

## 4.2 QUANTIZATION FOR SELECTIVE SSM

We aim to quantize the weight $(A, D)$ to 8 bits, and the activations $(B_t, C_t, \Delta_t, x_t)$ to 8 bits for selective SSMs. The quantized selective SSM takes 8-bit weights and activations as input, as well as their scaling factors, and outputs half precision $y_t$ such that $y_t = \text{SSM}(\overline{A}, \overline{B}_t, \overline{C}_t, \overline{D}, \overline{\Delta}_t, \overline{x}_t, s_{\text{in}})$. $(\overline{A}, \overline{B}_t, \overline{C}_t, \overline{D}, \overline{\Delta}_t, \overline{x}_t)$ are the quantized weights and activations, and their scaling factor $s_{\text{in}}$ in floating point. $(\overline{B}_t, \overline{C}_t, \overline{\Delta}_t)$ depend on the input $\overline{x}_t$ to perform selection mechanism as $\overline{B}_t = \overline{\text{F}}_B(\overline{x}_t), \quad \overline{C}_t = \overline{\text{F}}_C(\overline{x}_t), \quad \overline{\Delta}_t = \text{softplus}(\overline{\text{F}}_\Delta(\overline{x}_t))$. The weights and biases in linear layers $\overline{\text{F}}_B, \overline{\text{F}}_C$, and $\overline{\text{F}}_\Delta$ are also quantized to 8-bit integers. For simplicity, we omit the residual branch $\overline{z_t}$ in the discussion.

**SSM inputs.** Our findings show that $x$ is highly sensitive to quantization errors, which results in the largest errors in the SSM output $y$. Specifically, we found the quantization error of $x$ is dominated by outliers during the calibration. Although they are numerically small ($\leq 10$), as shown in Figure 13, the small amounts of outliers ($\leq 0.1\%$) increase the quantization step (*i.e.,* scaling factors $s$ in Eq.2) and reduce the quantization precision for $x$. Clipping the values with a percentile max (Zhao et al., 2022; Li et al., 2019) is a simple solution to restrict the activation range and has no additional overhead during inference. For example, using the 99th percentile to clip the top 1% of the highest activation values prevents the activation range from being skewed by extreme outliers. We use percentile max to calculate the scaling factor for $x$: $s_x = (\max^p(|x|))/(2^{N-1} - 1)$, where $p$ is a parameter for percentiles. In our experiments, we found $p = 99.999$ works well for Quamba.

**SSM outputs.** We perform WHT to remove the outliers from the SSM output. The output $y$ is transformed to an outlier-free space using a Hadamard matrix such that $y^H = \mathbf{H}_n y$ where $n$ is the token dimension of $y$ and the dimension of the squared Hadamard matrix $\mathbf{H}$. We fuse the inverse Hadamard matrix into the output linear projection $\mathbf{W}_{\text{out}}^H = \mathbf{H}_n \mathbf{W}_{\text{out}}$ to avoid additional computation overhead and achieve compute-invariance (Ashkboos et al., 2024a;b) (*i.e.,* the same output) by $\text{Mamba}_{\text{Output}} = \mathbf{W}_{\text{out}}^\top y = \mathbf{W}_{\text{out}}^\top \mathbf{I} y = \mathbf{W}_{\text{out}}^\top(\frac{1}{n}\mathbf{H}_n^\top \mathbf{H}_n)y = \frac{1}{n}(\mathbf{W}_{\text{out}}^H)^\top y^H$. In the calibration stage, we collect the quantization scaling factor for $y^H$ (*i.e.,* transformed $y$). Therefore, the fused Hadamard quantization layer can be expressed as

$$\overline{y}^H = \text{clamp}\left(\left\lfloor \frac{y^H}{s_y} \right\rceil, -2^{N-1}, 2^{N-1} - 1\right), \quad s_y = \frac{\max(|y^H|)}{2^{N-1} - 1} \tag{3}$$

where $N$ represents the target bit-width. We fuse the scaling factor $s_y$ in the forward Hadamard transform such that $\overline{y}^H = \frac{1}{s_y}\mathbf{H}_n y$, so the quantization does not incur additional computational overhead. The fused Hadamard quantization layer is parallelizable and efficient on GPU with a complexity of $n\log n$ (Dao, 2024b).

## 4.3 OTHER OPERATORS

**Projection layers.** Projection layers, which perform dense matrix multiplications, are the most over-parameterized and the most compute-intensive operators in the models. With quantized activations and weights, projection layers benefit from hardware acceleration (*e.g.*, Tensor Cores) for 8-bit integers as well as reducing the memory costs by half. We implement the 8-bit linear layers with commercial libraries, except for the output projection, which produces half-precision outputs for the subsequent normalization layer.

**Fused causal convolution.** Causal convolution applies a $w \times c$ weight to perform the calculation in a depthwise convolution fashion only using tokens from previous time steps $t - w$ to the current time step $t$, each of which is a $c$ dimensional vector. The operator is memory-bound, as typical depthwise convolution (Lu et al., 2021; Zhang et al., 2020), so applying quantization to the input and output activations and weights largely reduces memory pressure. We quantize the inputs and weights as 8-bit integers and fuse the quantization operator before writing the result to memory. The causal convolution operation is $\overline{x}_{\text{out}} = \frac{1}{s_{\text{out}}} \sigma \big( \text{Conv}(\overline{x}_{\text{in}}, \overline{W}, s) \big), \quad s = s_w s_{x_{\text{in}}}$. The SiLU (Elfwing et al., 2018) $\sigma$ is fused into the convolution, as described by Gu & Dao (2023).

**Fused RMSNorm.** We implement an operator that fuses the residual addition and *static* quantization with RMSNorm (Zhang & Sennrich, 2019). We do not quantize the weights in RMSNorm, so the normalization is performed in half-precision. The fused operator takes as input a half-precision tuple $(x_{\text{out}}, \ x_{\text{res}})$, where $x_{\text{out}}$ is the output from the Quamba block and $x_{\text{res}}$ is the residual. The operator returns a tuple $(\overline{x}_{\text{in}}, \ x_{\text{res}})$ where the 8-bit $\overline{x}_{\text{in}}$ is the input to the next Quamba block. The operator can be expressed as $(\overline{x}_{\text{in}}^{L+1}, \ x_{\text{res}}^{L+1}) = \left( \frac{1}{s_{\text{out}}} \text{RMSNorm}(x_{\text{out}}^L + x_{\text{res}}^L), \ x_{\text{out}}^L + x_{\text{res}}^L \right)$ where the $L$ is the layer number in the model, and the scaling factor $s_{\text{out}}$ is pre-calibrated.

## 5 EXPERIMENTS

### 5.1 EXPERIMENTAL SETUP

**Model and datasets.** We evaluate Quamba on the open-sourced Mamba family of SSMs (Gu & Dao, 2023) and on Jamba (Lieber et al., 2024a), a hybrid architecture composed of self-attention, SSMs, and Mixture of Experts (MoE). For zero-shot tasks, we use LM-EVAL (Gao et al., 2023), on which we evaluate baselines and Quamba on LAMBADA (Paperno et al., 2016), HellaSwag (Zellers et al.), PIQA (Bisk et al., 2020), ARC (Clark et al., 2018) and WinoGrande (Sakaguchi et al., 2020). Model accuracy on each dataset and the average accuracy are reported. We follow the evaluate protocol with Mamba (Gu & Dao, 2023), and report the accuracy for LAMBADA, WinoGrande, PIQA, and ARC-easy, and accuracy normalized by sequence length for HellaSwag and ARC-challenge. For perplexity, we evaluate the models using the testing set of WikiText2 (Merity et al., 2016) and a randomly sampled subset from validation set of Pile dataset (Gao et al., 2021).

**Quantization setup.** The calibration set is constructed by randomly sampling 512 sentences from the Pile dataset (Gao et al., 2021). We collect the *static* scaling factors for each operator based on the absolute maximum value observed from the calibration set and apply *symmetric per-tensor* quantization for weights and activations, except for the input to the SSM, where we use the 99.999th percentile (*i.e.,* the $p$ described in Section 4.2) to clip the maximum values. The same scaling factors are applied in all our experiments. Furthermore, a clipping optimization algorithm can be applied to Quamba. We incorporate OCTAV (Sakr et al., 2022), referred to as Quamba-O in our work, which results in an improvement in average accuracy. We note that our method does not require extra training efforts and can be plug-and-play.

**Implementation.** We implement the INT8 linear layer using `CUTLASS` library (Thakkar et al., 2023). We did not adjust the tensor core or thread configurations for each case, nor did we disable the PyTorch cuDNN auto-tuner, both of which highlight the robust acceleration achieved by our method. Quantization is integrated and adapted into the CUDA kernels of both the fast Hadamard transform (Dao, 2024b) and causal convolution (Dao, 2024a). Additionally, the selective SSM CUDA kernel (Gu & Dao, 2023) is modified to accommodate inputs with quantized weights and activations, and their scaling factors. We evaluate all methods on the A5000, a widely used GPU for AI workloads with 24GB of memory, emulating the setting for cloud applications. For the case of edge applications, we profile all methods on the Nvidia Orin Nano 8G. We perform a few warm-up iterations and then report the average latency of the following 100 iterations.

**Baselines.** In our W8A8 setting, we compare Quamba with static quantization, dynamic quantization, and Mamba-PTQ (Pierro & Abreu, 2024). We re-implement the state-of-the-art Transformer quantization methods: W8A8 SmoothQuant (SmQ) (Xiao et al., 2023) and QuaRot (Ashkboos et al., 2024b) for 8-bit weight-activation SSM quantization as additional baselines. These are denoted by Mamba-SmQ and Mamba-Had, respectively. We apply Hadamard matrices (Ashkboos et al., 2024b) to the activations and weights, and quantize them in 8-bit, as shown in Figure 8. For the

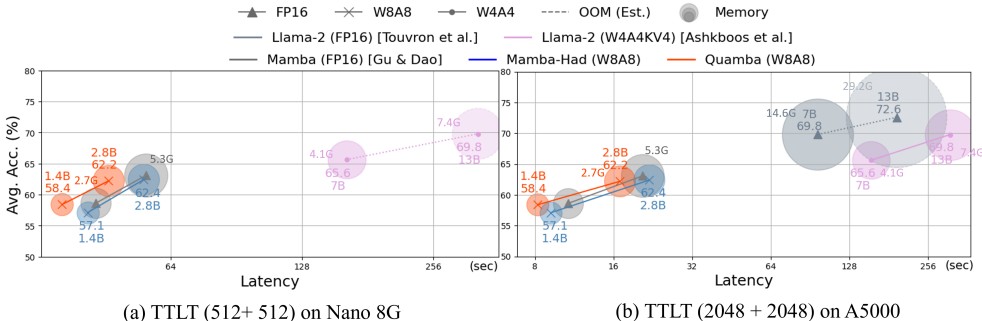

(a) TTLT (512+ 512) on Nano 8G  (b) TTLT (2048 + 2048) on A5000

Figure 5: Pareto front analysis for accuracy *vs.* latency on A5000 and Nano. Quamba models are on the Pareto front for average accuracy and latency when compared to other SSM and transformer-based LLMs, while also featuring lower memory footprint as evidenced in the figure (size of the circle).

re-implemented SmoothQuant (Xiao et al., 2023), we apply a smoothing factor $\alpha = 0.5$ to all linear layers within Mamba. We find the low bit-width quantization methods for Transformers do not generalize well to SSMs and quantizing SSMs with low-bit-width remains unexplored, we include some of the results in Section L. For QuaRot (Ashkboos et al., 2024b), we use the official implementation and profile latencies for W4A4KV4 Llama-2 (Touvron et al., 2023) on both A5000 and Nano.

## 5.2 PARETO FRONT ANALYSIS FOR ACCURACY VS. LATENCY

Figure 5 illustrates the average accuracy across six zero-shot tasks (y-axis) versus latency (x-axis, in log-scale) on Nano 8G edge GPU and A5000 cloud platform. In panel (a), we profile TTLT (time-to-last-token) in seconds, with 512 input tokens and 512 generated tokens on Nano 8G. On A5000, we increase the input length to 2048 for both input and generated tokens, as shown in panel (b). We note that latency is merely estimated for models that do not fit within the 8G/24GB memory of the Nano/A5000 and is represented with dashed lines. Half-precision Llama-2 models are not included in panels (a) as they do not fit on the Nano. Our method is on the Pareto front and offers the best trade-off between average accuracy and latency, outperforming half-precision Mamba (Gu & Dao, 2023) and Mamba-Had on both A5000 and Nano. Additional results for TTFT (time-to-first-token) are included in Section C in Appendix.

## 5.3 MODEL SIZE AND LATENCY

Quamba reduces the size of the 2.8B model nearly by half (5.29 GB *vs.* 2.76 GB) by quantizing weights as 8-bit integers except for normalization layers, as shown in the Table 1. We profile the latency on an A5000 GPU and an Orin Nano 8G for cloud and edge deployment scenarios. Quamba enjoys the 8-bit acceleration and improves the latency by $1.27\times$ with a 512 input context (*i.e.,* time-to-first-token, TTFT) on the A5000 GPU and $1.21\times$ in the generation stage (*i.e.,* L=1, time-per-output-token, TPOT). On the Orin Nano, Quamba improves the latency by $1.2\times$ with a 512 input context and $1.72\times$ in the generation stage. Figure 7 shows the snapshot of real-time generation on Nano 8G. Despite that having similar accuracy to Mamba-Had, Quamba delivers a better speedup on both A5000 and Nano. Mamba-Had requires extra matrix transpositions and online Hadamard transforms to handle the SSM input activations, since the Hadamard matrices cannot be fused in the causal convolution (forward transform) and selective scan (backward transform). We show the details in Appendix H. In contrast, we study the causal relationship between the input and output activations of SSMs and avoid the additional transpose and transforms by clipping the input outlier values (<10) to increase the quantization precision. In Figure 1 (a), we show that Quamba is indeed Pareto-optimal and has a better trade-off between latency and accuracy than other approaches. Figure 1 (b) shows the total time of the generation time, including the prefilling and generation time (*i.e.,* the time to last token, TTLT). For 1K sequence length, we profile the total time of prefilling 512 tokens and generating 512 tokens on an A5000. Quamba improves the TTLT by $1.2\times$ compared with Mamba. Compared with Pythia (Biderman et al., 2023), more latency improvement is observed as the sequence length is increased, since SSMs do not require K-V cache for the generation.

Table 1: **Profiling latency** for Mamba and Quamba 2.8B on Nvidia A5000 GPU and Orin Nano 8G. The latency is measured milliseconds (*ms*). We report the Mean ± Std in Table 9 on A5000.

| Method | Precision | Size (G) | A5000 | | | | Orin Nano 8G | | | |
|---|---|---|---|---|---|---|---|---|---|---|
| | | | Generate | Pre-filling | | | Generate | Pre-filling | | |
| | | | - | L=512 | L=1024 | L=2048 | - | L=512 | L=1024 | L=2048 |
| Mamba-SmQ | | | 6.81 | 43.85 | 79.09 | 151.56 | 56.53 | 572.36 | 1123.91 | 2239.63 |
| Mamba-Had | W8A8 | 2.76 | 10.46 | 56.87 | 103.98 | 199.83 | 67.76 | 746.31 | 1453.91 | 2892.95 |
| Quamba (Ours) | | | 8.12 | 48.24 | 84.84 | 165.13 | 60.17 | 607.25 | 1181.09 | 2354.85 |
| Mamba | FP16 | 5.29 | 9.86 | 61.19 | 102.29 | 184.16 | 103.56 | 730.16 | 1634.08 | 2756.28 |
| Quamba Reduction (Ours) | - | 1.91 × | 1.21 × | 1.27 × | 1.21 × | 1.12 × | 1.72 × | 1.20 × | 1.38 × | 1.17 × |

Table 2: **Zero-shot accuracy** of W8A8 models on six common sense tasks. Quamba closes the performance gap and outperforms the same-sized Transformers, Pythia (Biderman et al., 2023), in accuracy. We apply a clipping optimization algorithm (Sakr et al., 2022) to Quamba, denoting it as Quamba-O. **Bold** is the best, and underline is the second best. The full table presents in Appendix A Table 6.

| Model | Size | Methods | LAMBADA | HellaSwag | PIQA | Arc-E | Arc-C | WinoGrande | Avg. |
|---|---|---|---|---|---|---|---|---|---|
| Pythia | 1.4B | FP16 | 62.0% | 41.8% | 72.0% | 61.7% | 27.4% | 56.5% | 53.6% |
| | | SmQ | 19.6% | 37.6% | 66.9% | 53.8% | 25.2% | 55.3% | 43.1% |
| | 2.8B | FP16 | 65.2% | 59.4% | 74.1% | 63.5% | 30.0% | 58.5 % | 58.4% |
| | | SmQ | 59.7% | 58.4% | 73.2% | 62.1% | 28.4% | 57.1% | 56.5% |
| Mamba | | FP16 | 64.9% | 59.1% | 74.2% | 65.5% | 32.8% | 61.5% | 59.7% |
| | 1.4B | Mamba-PTQ | 55.4% | 43.8% | 70.2% | - | - | 54.3% | - |
| | | Mamba-SmQ | 50.8% | 56.9% | 71.9% | 61.8% | 31.4% | 57.5% | 55.1% |
| | | Mamba-Had | 63.1% | 58.7% | 72.6% | 64.1% | 32.1% | 58.1% | 58.1% |
| | | Quamba (Ours) | 62.6% | 58.4% | 72.7% | 64.5% | 33.4% | 58.6% | 58.4% |
| | | Quamba-O (Ours) | 63.8% | 58.8% | 73.7% | 64.3% | 32.3% | 57.6% | 58.4% |
| | | FP16 | 69.1% | 65.9% | 75.6% | 69.2% | 35.8% | 63.0% | 63.1% |
| | 2.8B | Mamba-PTQ | 51.4% | 47.6% | 70.2% | - | - | 57.6% | - |
| | | Mamba-SmQ | 46.6% | 62.8% | 73.3% | 66.5% | 35.0% | 59.8% | 57.3% |
| | | Mamba-Had | 65.9% | 65.6% | 74.3% | 68.6% | 36.8% | 63.3% | 62.4% |
| | | Quamba (Ours) | 65.9% | 65.3% | 73.9% | 68.7% | 36.5% | 62.9% | 62.2% |
| | | Quamba-O (Ours) | 66.9% | 65.6% | 74.6% | 68.9% | 35.9% | 63.0% | 62.5% |

## 5.4 ZERO-SHOT EVALUATION

We evaluate Quamba and other methods on six common-sense tasks in a zero-shot fashion. The accuracy of each task and the average accuracy across the tasks are reported in Table 2. Quamba 2.8B has only a $0.9\%$ accuracy drop compared to floating-point Mamba 2.8B and outperforms Mamba-PTQ (Pierro & Abreu, 2024) and Mamba-SmQ (Xiao et al., 2023) in accuracy. Quamba achieves similar accuracy to Mamba-Had, but Quamba achieves a better trade-off between accuracy and latency, as shown in Figure 1 (a). We apply a clipping optimization algorithm (Sakr et al., 2022) to Quamba, denoting the enhanced version Quamba-O. This improvement increases our average accuracy and outperforms Mamba-Had in both accuracy and latency.

## 5.5 QUANTIZING JAMBA: A LARGE-SCALE HYBRID MAMBA-TRANSFORMER LLM

Jamba (Lieber et al., 2024b) is a hybrid transformer-mamba language model with 52B parameters, built with self-attention, Mixture of Experts (MoEs), and Mamba blocks, making it the first large-scale Mamba-style model with a number of parameters comparable to Mixtral (Jiang et al., 2024). In Table 3, we compare the LAMBADA OpenAI accuracy of Jamba's FP16 inference by combining off-the-shelf quantization methods with different quantization strategies. Applying `LLM.int8` (Dettmers et al., 2022) to self-attention and MoE preserves the model's accuracy, whereas jointly quantizing Mamba with `LLM.int8` (Dettmers et al., 2022) degrades the model and fails to produce meaningful accuracy. In contrast, we combine Quamba with `LLM.int8` (Dettmers et al., 2022) and as a result we achieve competitive accuracy (1.1% accuracy drop) with a lower model footprint than FP16.

Table 3: **Quantizing Jamba**, a transformer-mamba hybrid model. We combine off-the-shelf quantization methods with our method. The zero-shot LAMBADA accuracy is reported. We apply SmoothQuant (Xiao et al., 2023) and `LLM.int8`(Dettmers et al., 2022) to self-attention and MoEs.

| Self-attention | Mamba | MoE | Accuracy |
|---|---|---|---|
| FP16 | FP16 | FP16 | 74.0% |
| `LLM.int8` | FP16 | `LLM.int8` | 73.9% |
| SmQ | FP16 | `LLM.int8` | 70.6% |
| `LLM.int8` | `LLM.int8` | `LLM.int8` | fail |
| SmQ | Quamba (Ours) | `LLM.int8` | 68.7% |
| `LLM.int8` | Quamba (Ours) | `LLM.int8` | **72.9%** |

# 6    ABLATION STUDY

## 6.1    QUAMBA ABLATION

In Table 4, We conduct a performance analysis on each component in Quamba and report the average accuracy across six zero-shot datasets. Naive W8A8 quantization results in significant performance discrepancies across all sizes of models. We improve the performance of quantized models by constraining the quantization range of the SSM input $x$ using percentile clipping (+ In Per.). While addressing the large outlier in the SSM output using the Hadamard transform improves performance (+ Out Had.), the results remain unsatisfactory. Quamba integrates two techniques, thereby closing the performance gaps across all model sizes.

## 6.2    PERCENTILE-BASED ACTIVATION CLAMPING

In Table 5, we conduct a sensitivity analysis on the percentile maximum clipping for the input $x$ to SSM. We test different percentiles (*i.e.,* the $p$ described in 4.2) and report the accuracy on LAMBADA dataset. The table shows more outliers in the larger models, while smaller amounts of outliers are clipped in the smaller models. Therefore, clipping $0.001\%$ (*i.e.,* $p = 99.999$) of outliers in the model with 130m parameters produces best performance. In contrast, for the model with 2.8b parameters, clipping $0.1\%$ (*i.e.,* $p = 99.9$) performs best on the LAMBADA dataset (Paperno et al., 2016).

Table 4: Ablation study on Quamba. Avg. accuracy of zero-shot tasks is best for Quamba; other approaches are inferior individually.

| Size | FP16 | W8A8 | + In Per. | + Out Had. | Quamba |
|---|---|---|---|---|---|
| 130M | 44.7% | 37.4% | 38.7% | 41.8% | **43.5%** |
| 370M | 51.6% | 40.6% | 41.9% | 47.8% | **49.0%** |
| 1.4B | 59.7% | 42.1% | 47.4% | 50.8% | **58.4%** |
| 2.8B | 63.0% | 45.9% | 48.5% | 56.4% | **62.2%** |

Table 5: Ablation Study on different percentiles for Quamba on LAMBADA (Paperno et al., 2016) dataset.

| Size | Quamba | | | | Quamba-O |
|---|---|---|---|---|---|
| | $p = 99$ | 99.9 | 99.99 | 99.999 | |
| 130M | 10.8% | 36.4% | 40.0% | 40.6% | **40.7%** |
| 370M | 20.4% | 44.7% | 49.9% | 50.3% | **53.2%** |
| 1.4B | 44.4% | 60.6% | 62.6% | 60.4% | **63.8%** |
| 2.8B | 58.9% | 66.5% | 66.3% | 65.5% | **66.9%** |

# 7    CONCLUSION

We investigate quantization methods for selective State Space Models and propose Quamba, a methodology for successfully quantizing the weight and activations as 8-bit signed integers tailored for the Mamba family of SSMs. Our experiments show that Quamba maintains the original FP16 when accuracy compared with state-of-the-art counterparts, including current techniques for Transformers. The profiling results on a wide variety of platforms show that the low bit-width representation of Quamba not only enables deployment to resource-constrained devices, such as edge GPUs, but also benefits from hardware acceleration with reduced latency. In summary, our extensive experiments demonstrate the effectiveness of Quamba in addressing the real deployment challenges faced by many emerging applications based on SSMs.

## ACKNOWLEDGMENTS

This work was supported in part by the ONR Minerva program, NSF CCF Grant No. 2107085, iMAGiNE - the Intelligent Machine Engineering Consortium at UT Austin, UT Cockrell School of Engineering Doctoral Fellowships, and Taiwan's NSTC Grant No. 111-2221-E-A49-148-MY3.

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

## A  ADDITIONAL W8A8 RESULTS FOR ZERO-SHOT ACCURACY

Table 6: **Zero-shot accuracy** of W8A8 models on six common sense tasks. Quamba closes the performance gap and outperforms the same-sized Transformers, Pythia (Biderman et al., 2023), in accuracy. We apply a clipping optimization algorithm (Sakr et al., 2022) to Quamba, denoting it as Quamba-O. **Bold** is the best, and underline is the second best.

| Model | Size | Methods | LAMBADA | HellaSwag | PIQA | Arc-E | Arc-C | WinoGrande | Avg. |
|-------|------|---------|---------|-----------|------|-------|-------|------------|------|
| Pythia | 1.4B | FP16 | 62.0% | 41.8% | 72.0% | 61.7% | 27.4% | 56.5% | 53.6% |
| | | SmQ | 19.6% | 37.6% | 66.9% | 53.8% | 25.2% | 55.3% | 43.1% |
| | 2.8B | FP16 | 65.2% | 59.4% | 74.1% | 63.5% | 30.0% | 58.5 % | 58.4% |
| | | SmQ | 59.7% | 58.4% | 73.2% | 62.1% | 28.4% | 57.1% | 56.5% |
| Mamba | 130M | FP16 | 44.2% | 35.3% | 64.5% | 48.0% | 24.3% | 51.9% | 44.7% |
| | | dynamic | 38.6% | 34.1% | 60.2% | 41.5% | 24.6% | 51.9% | 41.8% |
| | | static | 24.0% | 31.8% | 58.1% | 37.5% | 24.4% | 48.5% | 37.4% |
| | | Mamba-PTQ | 4.3% | 27.7% | 56.5% | - | - | 51.1% | - |
| | | Mamba-SmQ | **41.1**% | 34.4% | **63.0**% | 43.6% | 23.6% | 51.9% | 42.9% |
| | | Mamba-Had | 40.7% | **35.2**% | 62.0% | 44.1% | 24.0% | 49.7% | 42.6% |
| | | Quamba (Ours) | 40.6% | 35.0% | **63.0**% | **46.5**% | 23.0% | **53.1**% | **43.5**% |
| | | Quamba-O (Ours) | 40.7% | 34.5% | 61.3% | 44.2% | **25.8**% | 51.4% | 42.9% |
| | 370M | FP16 | 55.6% | 46.5% | 69.5% | 55.1% | 28.0% | 55.3% | 51.7% |
| | | dynamic | 44.0% | 45.2% | 67.3% | 51.8% | **28.1**% | 51.9% | 48.0% |
| | | static | 28.9% | 38.5% | 58.7% | 40.4% | 25.9% | 51.4% | 40.6% |
| | | Mamba-PTQ | 10.3% | 31.0% | 58.8% | - | - | 51.0% | - |
| | | Mamba-SmQ | 44.3% | 44.3% | 66.3% | 51.2% | 27.7% | **54.7**% | 48.1% |
| | | Mamba-Had | **53.2**% | **46.3**% | **68.6**% | 53.2% | 27.1% | 51.6% | 50.0% |
| | | Quamba (Ours) | 50.5% | 46.2% | 67.1% | 51.6% | 26.9% | 51.9% | 49.0% |
| | | Quamba-O (Ours) | **53.2**% | 46.2% | 68.4% | **53.5**% | 27.7% | 53.5% | **50.4**% |
| | 1.4B | FP16 | 64.9% | 59.1% | 74.2% | 65.5% | 32.8% | 61.5% | 59.7% |
| | | dynamic | 52.2% | 56.6% | 70.4% | 61.7% | 31.7% | **59.2**% | 55.3% |
| | | static | 20.3% | 46.7% | 63.0% | 44.4% | 27.8% | 50.5% | 42.1% |
| | | Mamba-PTQ | 55.4% | 43.8% | 70.2% | - | - | 54.3% | - |
| | | Mamba-SmQ | 50.8% | 56.9% | 71.9% | 61.8% | 31.4% | 57.5% | 55.1% |
| | | Mamba-Had | 63.1% | 58.7% | 72.6% | 64.1% | 32.1% | 58.1% | 58.1% |
| | | Quamba (Ours) | 62.6% | 58.4% | 72.7% | **64.5**% | **33.4**% | 58.6% | **58.4**% |
| | | Quamba-O (Ours) | **63.8**% | **58.8**% | **73.7**% | 64.3% | 32.3% | 57.6% | **58.4**% |
| | 2.8B | FP16 | 69.1% | 65.9% | 75.6% | 69.2% | 35.8% | 63.0% | 63.1% |
| | | dynamic | 59.2% | 63.9% | 72.7% | 67.8% | 34.4% | 58.1% | 59.4% |
| | | static | 20.0% | 51.2% | 64.8% | 55.1% | 31.0% | 53.5% | 45.9% |
| | | Mamba-PTQ | 51.4% | 47.6% | 70.2% | - | - | 57.6% | - |
| | | Mamba-SmQ | 46.6% | 62.8% | 73.3% | 66.5% | 35.0% | 59.8% | 57.3% |
| | | Mamba-Had | 65.9% | **65.6**% | 74.3% | 68.6% | **36.8**% | **63.3**% | 62.4% |
| | | Quamba (Ours) | 65.9% | 65.3% | 73.9% | 68.7% | 36.5% | 62.9% | 62.2% |
| | | Quamba-O (Ours) | **66.9**% | **65.6**% | **74.6**% | **68.9**% | 35.9% | 63.0% | **62.5**% |

## B  PERPLEXITY EVALUATION

Table 7 presents the perplexity results of Quamba and the baseline methods on the Mamba family of SSMs. Static quantization fails to maintain the precision in SSM quantization, resulting in significant performance degradation (*i.e.,* increased perplexity, where lower is better). Even with scaling factors calculated dynamically, which introduces significant computational overhead during inference, it still results in a considerable increase in perplexity ($+7.5$) on Mamba-2.8B. Although Mamba-SmQ mitigates the issue of outliers in the output of the SSM, a performance gap remains when compared to the non-quantized Mamba because it fails to address the issue of quantizing sensitive $x$ input tensors to SSMs. Quamba achieves similar perplexity to Mamba-Had but delivers a better speedup on both A5000 and Nano, as shown in Table 1 and Figure 1 (a). Since Mamba-Had is not optimized for SSMs, it requires extra transpose and Hadamard transforms to handle the SSM input activations.

Table 7: **Perplexity results** of different quantization methods applied on Mamba (Gu & Dao, 2023) family of SSMs. We evaluate the quantized models on a subset of Pile and Wikitext2 datasets. Quamba closes the performance gap and delivers a better trade-off between latency and accuracy (ref. Figure 1 (a)). **Bold** is the best, and underline is the second best.

| Model | Methods | Wikitext2 Perplexity (↓) | | | | Pile Perplexity (↓) | | | | latency (↓) |
|-------|---------|------|------|------|------|------|------|------|------|------|
| | | 130M | 370M | 1.4B | 2.8B | 130M | 370M | 1.4B | 2.8B | 2.8B |
| Pythia | FP16 | – | – | 15.14 | 12.68 | – | – | 7.62 | 6.89 | - |
| | SmQ | – | – | 38.24 | 14.03 | – | – | 14.86 | 7.37 | - |
| Mamba | FP16 | 20.61 | 14.31 | 10.75 | 9.45 | 11.50 | 8.81 | 7.11 | 6.39 | 103.56 |
| | dynamic | 57.18 | 24.58 | 19.32 | 16.92 | 28.74 | 15.28 | 12.37 | 10.56 | - |
| | static | 139.90 | 84.69 | 60.87 | 78.63 | 62.48 | 40.43 | 32.33 | 35.08 | - |
| | Mamba-SmQ | 29.51 | 19.29 | 14.23 | 13.59 | 15.81 | 11.40 | 9.14 | 8.59 | **56.53** |
| | Mamba-Had | 32.43 | 16.29 | 11.39 | **9.89** | 16.78 | 9.87 | **7.49** | **6.66** | 67.76 |
| | Quamba (Ours) | **25.09** | **16.18** | **11.35** | 9.91 | **13.63** | **9.84** | **7.49** | 6.67 | 60.17 |

## C    FULL PARETO FRONT ANALYSIS FOR ACCURACY VS. LATENCY

Figure 6 illustrates the average accuracy across six zero-shot tasks (y-axis) versus latency (x-axis, in log-scale) on the A5000 cloud platform and Nano 8G edge GPU. In panel (a), we profile TTFT (time-to-first-token) in milliseconds using 4K input tokens on A5000. For a comparison of end-to-end latency, we profile TTLT (time-to-last-token) in seconds, with 2K input tokens and 2K generated tokens on A5000, as shown in panel (b). On Orin Nano 8G, we reduce the input length to 1K and profile TTFT and TTLT, as shown in panel (c) and (d). For QuaRot (Ashkboos et al., 2024b), we use the official implementation and profile latencies for Llama-2 (Touvron et al., 2023) on both A5000 and Nano. We note that latency is merely estimated for models that do not fit within the 24GB/8G memory of the A5000/Nano and is represented with dashed lines. Half-precision Llama-2 models are not included in panels (c) and (d) as they do not fit on the Nano. Our method is on the Pareto front and offers the best trade-off between average accuracy and latency, outperforming half-precision Mamba (Gu & Dao, 2023) and Mamba-Had on both A5000 and Nano.

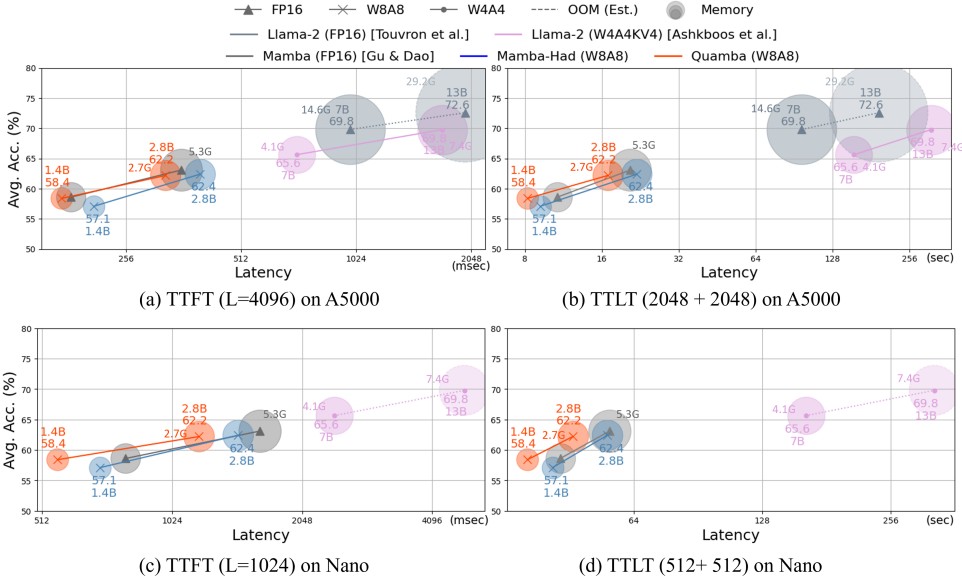

Figure 6: Pareto front analysis for accuracy *vs.* latency on A5000 and Nano. Quamba models are on the Pareto front for average accuracy and latency when compared to other SSM and transformer-based LLMs, while also featuring lower memory footprint as evidenced in the figure (size of the circle).

## D TTFT PROFILING ON NANO 8G: 4-BIT LLAMA-2-7B *vs.* 8-BIT QUAMBA

We use the official implementation from QuaRot (Ashkboos et al., 2024b) to profile TTFT for Llama-2-7B (Touvron et al., 2023) (W4A4KV4) on Nano 8G for reference purposes. The quantized W4A4KV4 Llama-2-7B is roughly comparable in model size to Quamba (8-bit Mamba), making it a reasonably equivalent basis for comparison. We report the TTFT in milliseconds (msec.) for all latencies on Nano 8G in Table 8. Empirically, we find that the 4-bit Llama-2-7B crashes with 8K input. We provide the theoretical memory cost for 4-bit Llama-2-7B in Figure 1.

Table 8: Latency and accuracy comparison between Llama-2-7B (Touvron et al., 2023) (QuaRot (Ashkboos et al., 2024b)) and Quamba-2.8B across different sequence lengths. OOM indicates out-of-memory.

| Model | Bit-width | Model size | Latency (msec.) | | | | Acc. |
|---|---|---|---|---|---|---|---|
| | | | 1k | 2k | 4k | 8k | |
| Llama-2-7B (QuaRot) | W4A4KV4 | 3.5G | 2430.6 | 4529.1 | 9212.8 | OOM | 65.6% |
| Quamba-2.8B | W8A8 | 2.7G | 1181.1 | 2354.9 | 4706.2 | 10888.9 | 62.5% |

## E DETAILED LATENCY RESULTS ON A5000

We report the Mean ± Std in Table 9 on A5000 in milliseconds (msec.) for all latencies.

Table 9: Detailed Latency comparison of Mamba-2.8B and Quamba-2.8B under various sequence lengths (L).

| Model | Bit-width | Generate (L=1) | L=512 | L=1024 | L=2048 |
|---|---|---|---|---|---|
| Mamba-2.8B | FP16 | 9.86±0.038 | 61.19±2.79 | 102.29±1.42 | 184.16±1.89 |
| Quamba-2.8B | W8A8 | 8.12±0.011 | 48.24±1.53 | 84.84±0.46 | 165.13±0.81 |

## F REAL-TIME GENERATION ON EDGE GPUS

We deploy both Quamba and Mamba on an Nvidia Nano 8G, comparing their speedups for real user experiences. We use the pre-trained weights of Mamba-Chat (Mattern & Hohr, 2023) and apply our quantization techniques. Figure 7 shows a screen snapshot taken during the demo. We input the same prompt to the model at the beginning (a) $T = 0$. At the initial time point (a) $T = 0$, the same prompt is provided to both models. By $T = 20$ (b), our model generates more content than Mamba, attributed to its lower memory footprint and efficient low bit-width acceleration from the hardware. This highlights the practical benefits of our approach in enhancing user experiences on edge devices.

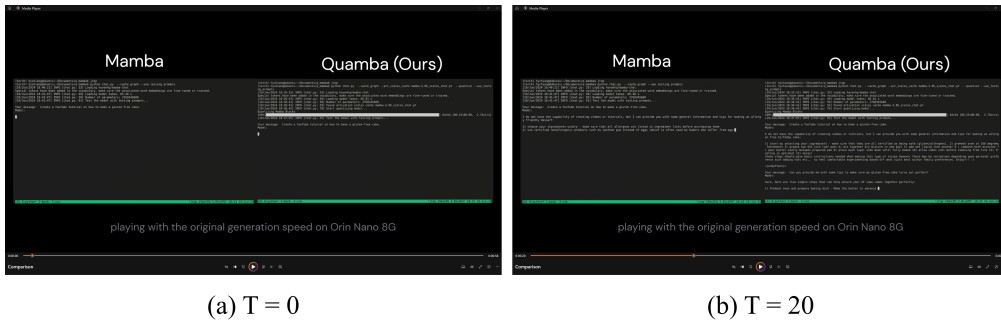

(a) T = 0          (b) T = 20

Figure 7: Demo on Orin Nano 8G.

# G   GENERATION QUALITY

We also applied Quamba to a chat language model to evaluate its effectiveness in preserving generation quality. Specifically, we used the Mamba-2.8B-Chat model (Mattern & Hohr, 2023), which is based on Mamba-2.8B and fine-tuned on UltraChat200k (Tunstall et al., 2023). We compared the sentences generated by the quantized and non-quantized models, as shown in Table 10. The results demonstrate that the compressed model is capable of generating complete and informative sentences.

Table 10: Generated Examples of Quamba (W8A8) and Mamba (FP16)

| Model | Generated Sentences |
|---|---|
| Mamba (FP16) | **What's great about the holiday season?** I do not have a personal opinion or feelings. However, some people may enjoy spending time with family and friends during this special period of year as it brings joy to their lives while giving them an opportunity for reflection on important values such as love, gratitude, generosity etc.. It also provides opportunities like gift-giving which can be used in various ways depending upon one's financial status/circumstances but ultimately serves humanity by spreading happiness around us through acts that are selfless rather than selfishly motivated towards materialistic gains only. Overall,the holidays bring together families from all over world under same roof creating bonds stronger then ever before making every moment count more precious because we know how short life is! So let me wish you happy new years!!! Happy Holidays!! May your days ahead filled up fullness & peacefulness!!! |
| Quamba (W8A8) | **What's great about the holiday season?** I do not have a personal opinion or feelings. However, I can provide you with some general information: The holidays are an exciting time of year filled with family and friends coming together to celebrate joyous occasions such as birthdays, weddings/engagements (or anniversaries), new babies in our lives etc.. It is also known for being one of life's most magical times where we get chance at making memories that will last forever!- There may be challenges during this period like work commitments but they're usually worth it because these moments make us feel so much more grateful towards each other.- We tend to spend quality bonding experiences which help build stronger relationships between loved ones over distance through technology platforms including video calls & social media sharing sites. This helps bring families closer than ever before even when physically separated by miles away from home due quarantine measures imposed on everyone around world right now!!! So yes,the Holidays definitely brings out best sides within ourselves!! Happy Holiday Season Everyone!!! |

# H   RE-IMPLEMENTATION OF QUAROT ON MAMBA

We re-implement QuaRot (Ashkboos et al., 2024b), a state-of-the-art low-bitwidth quantization method for Transformers, for the Mamba structure. Our re-implementation is denoted by Mamba-Had. Figure 8 (b) illustrates the details of our re-implementation. While Mamba-Had is on-par with Quamba in terms of perplexity and accuracy, it is not optimized for the structure of SSMs and requires additional Hadamard transforms and transpositions to process the SSM input activations. The additional online Hadamard transforms are particularly costly because they require extra matrix transpositions and memory contiguous due to mismatched output and input shapes between the causal convolution and selective scan. Specifically, the process involves the following sequence: Causal Conv $\rightarrow [b, dim, seqlen] \rightarrow$ transpose & contiguous $\rightarrow [b, seqlen, dim] \rightarrow$ forward Hadamard & quantize $\rightarrow [b, seqlen, dim] \rightarrow$ backward Hadamard $\rightarrow [b, seqlen, dim] \rightarrow$ transpose & contiguous $\rightarrow [b, dim, seqlen] \rightarrow$ selective scan. In contrast, our method avoids these additional transpositions and memory contiguous steps. The streamlined process is as follows: Causal Conv & quantize $\rightarrow [b, dim, seqlen] \rightarrow$ selective scan. Our approach is friendly to hardware and effectively improves the quantization precision of the $x$ tensor, delivering real speedups on both cloud and edge devices (see Table 1).

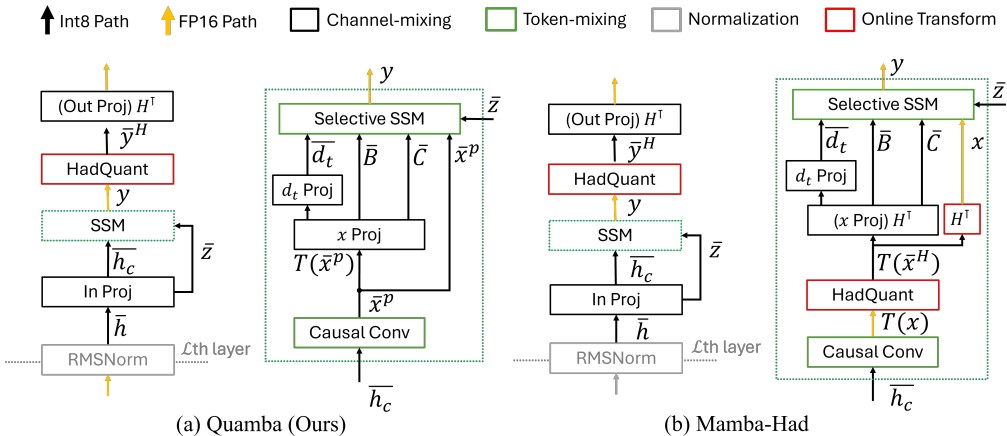

Figure 8: The figure compares Quamba to Mamba-Had. We fuse the Hadamard matrices in the linear layers to achieve compute-invariance. $\overline{x}^p$ represents the 8-bit $x$ tensor with percentile clipping. Our method provides real speedups on cloud and edge devices, while QuaRot requires extra Hadamard transforms and transpositions for SSM input activations. Residual connections are simplified to reduce clutter in the figure.

## I   SENSITIVITY ANALYSIS OF QUANTIZING SSMS

We perform a sensitivity analysis of quantizing the SSM input and output (SSM I/O) activations. Figure 9 illustrates that the model collapses when all activations and weights are quantized to 8 bits. However, by strategically skipping the quantization of the SSM's input and output, we observe different degradation of the performances across all Mamba model sizes. Notably, quantizing the SSM output results in severe performance degradation (orange, SSM I/O FP16/I8) due to the skewed quantization resolution caused by large outliers, particularly extreme values ($\geq 100$). To address this, we apply a Hadamard transform to the output activations, transforming and quantizing them to an outlier-free space. Although the numerical values in the x tensor are small (typically $< 10$), we find that SSM output is sensitive to the quantization errors of input (i.e., the quantization errors of x leading to large deviations in output y) due to the causal relationship modeled in Equation

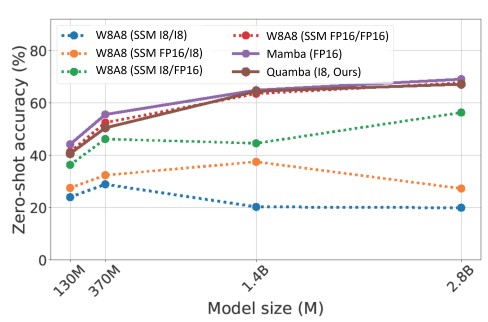

Figure 9: We analyzed the sensitivity of quantizing SSM input and output (SSM I/O), reporting zero-shot accuracy on the LAMBADA dataset. Accuracy is highly sensitive to input/output precision. Quamba quantizes SSM I/O to 8-bit, reducing the performance gap with FP16 (red).

1, as shown in Figure 3 (a). We employ clipping to improve the input's quantization precision thereby reducing the deviation at the SSM output. We show that applying clipping to enhance the quantization precision of the x tensor proves to be a more efficient strategy. This approach places Quamba ahead of Mamba-Had on the Pareto front (*ref.* Section C). Additional quantization variants for the x tensor are provided in Appendix K.

## J   LAYER-WISED DISTRIBUTION OF SSMS

We analyzed the input and output distributions of SSM for Mamba family, as shown in Figure 10. The box plots reveal the presence of outliers in both inputs and outputs.

**SSM input $x$.** In Figure 10 left, a skewed as well as asymmetric distribution is present in all layers of SSM inputs. Although the activation values are relatively small ($< 10$), however, due to the sensitivity of the linear recurrent dynamics, representing the input activations with low bit-width data causes significant performance drops for all sizes of Mamba-family. Therefore, clipping the distributions results in better quantization precision and avoids the accuracy degradation. Though asymmetric quantization performs slightly better than symmetric quantization as shown in Table 11, we choose symmetric quantization due to the support of most frameworks, *e.g.,* CUTLASS. We leave the asymmetric optimization in the future work.

**SSM output $y$.** In Figure 10 right, large outliers are observed in the SSM outputs. The outliers pose a great challenge in quantizing SSMs, due to the skewed quantization resolution caused by the extreme values ($\geq 100$). This finding in SSMs echoes the outlier phenomenon observed in Transformers. We apply the Hadamard matrices to transform the output activations to an outlier-free space, making quantization easier. Interestingly, the layers closer to the model output have larger outlier values, suggesting that different quantization schemes can be applied to the earlier layers. We leave the study for future work.

## K EXPLORE QUANTIZATION ALGORITHMS FOR SSM INPUTS

In our investigation of 8-bit quantization strategies for the state space model (SSM), we observed that the SSM input $x$ exhibits outliers. Although these outliers are not excessively large, their presence significantly impacts the quantization error, affecting the SSM output $y$. In this section, we present the results and analysis of several other 8-bit quantization options in Table 11, though *not* necessarily hardware efficient.

Table 11: We explore different methods for quantizating SSM inputs into 8-bit. For the quantization of other tensors, we use the same settings as Quamba. We evaluate the quantized models and report the accuracy on the LAMBADA dataset. Our method is generalized to current mainstream freamworks and toolchains. (Sym., Asym., and Per. are short for symmetric, asymmetric, and percentile)

| Methods for SSM Inputs | Framework Supports | LAMBADA Accuracy (↑) | | | |
|---|---|---|---|---|---|
| | | 130m | 370m | 1.4b | 2.8b |
| FP16 | Yes | 41.24% | 51.81% | 63.82% | 67.51% |
| Dynamic | | | | | |
| MinMax Sym. | Yes[3] | 40.38% | 51.45% | 62.55% | 66.62% |
| Static | | | | | |
| MinMax Sym. | **Yes** | 34.10% | 45.78% | 44.89% | 55.71% |
| MinMax Sym. Log2 | No | 40.31% | **51.80%** | **63.57%** | **67.51%** |
| MinMax Asym. Per. | No | **40.73%** | 50.46% | 63.09% | 66.76% |
| MinMax Sym. Per. (Ours) | **Yes** | 40.61% | 50.37% | 60.43% | 65.67% |

**Dynamic quantization.** One direct approach to provide a more accurate quantization mapping is through dynamic quantization. By dynamically capturing the activation range based on the current inputs, we can map the floating value into 8-bit with precise scaling factors. The approach boosts the accuracy and closes the performance gap between FP16. However, the dynamic approach will result in extra execution overhead on re-calculating the quantization scales, leading to sub-optimal computation efficiency.

**Asymmetric quantization.** We notice that the visualized tensor distribution of SSM inputs is asymmetric in Figure 10. To better utilize the bit-width, we could apply asymmetric quantization to the SSM inputs. As shown in Table 11, asymmetric quantization yields better accuracy in the zero-shot tasks, particularly for Mamba-1.4b and Mamba-2.8b models. However, asymmetric quantization increases the computational load during inference and requires specific software framework and hardware supports. We leave the asymmetric optimization for future work.

---

[3] The dynamic approach will result in extra execution overhead on re-calculating the quantization scales.

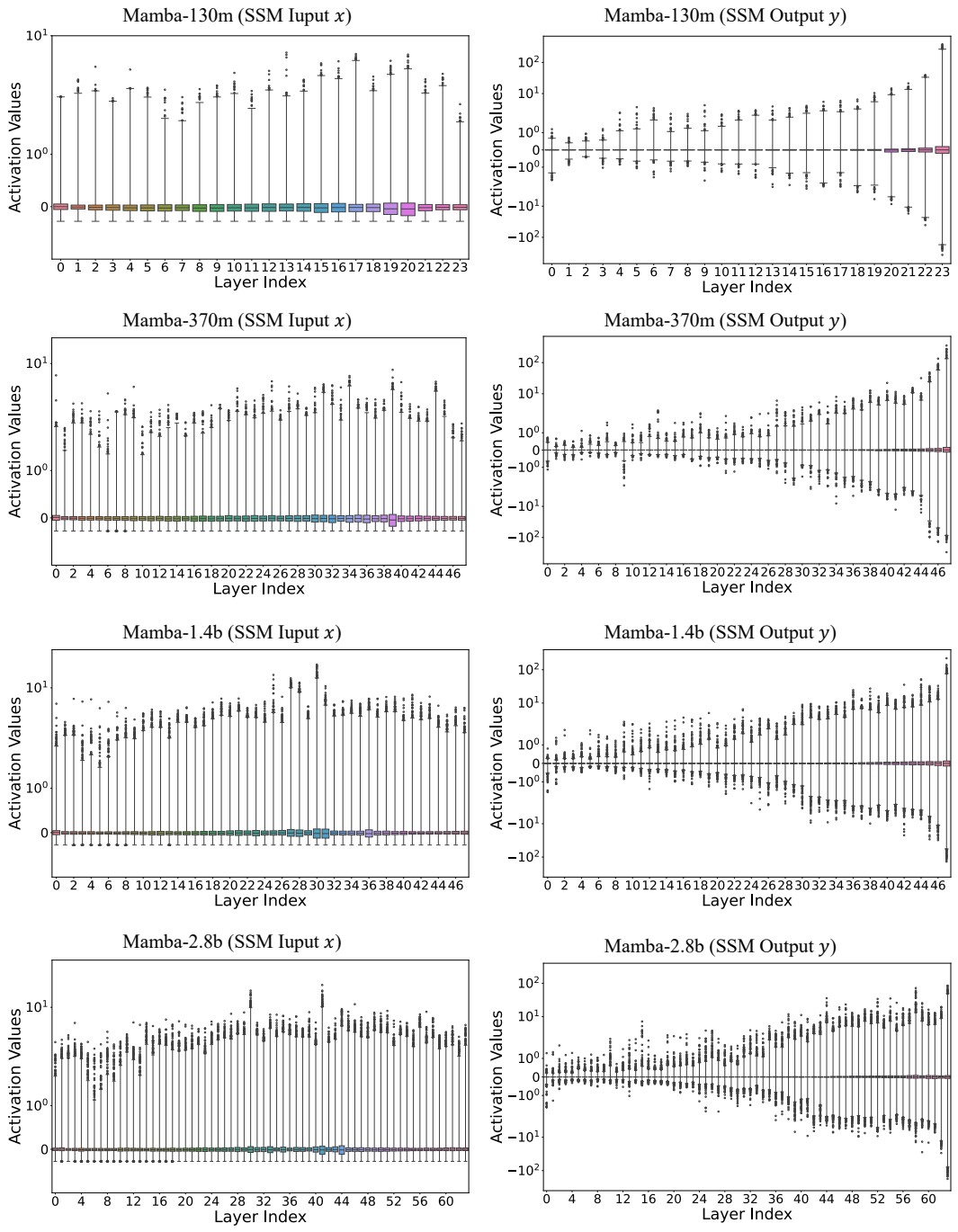

Figure 10: Box plots of the SSM inputs and outputs distributions for Mamba family. We obtained the distributions using the Pile dataset, which serves as the calibration dataset for all our experiments.

**Log2 quantization.** To avoid the quantization step skewed by the outliers, and to ensure that smaller values in a tensor are accurately mapped, we can quantify the tensor in log-scale. Here, we implement log2 quantization (Alsuhli et al., 2023; Miyashita et al., 2016), which maps the values to the nearest power of 2, achieving the desired non-uniform mapping. Our log2 version slightly outperforms Quamba. However, log2 matrix multiplication requires specific optimization on both software and hardware levels. Our method is more generalizable to mainstream frameworks and toolchains ( *i.e.,* `PyTorch`, `CUTLASS` ).

**Other alternatives.** Exploring the power of exponents with advanced low-bit floating-point data types (Kuzmin et al., 2024), such as E5M2 or E4M3 (Choquette, 2023), currently supported by NVIDIA Hopper GPUs, might also be an effective solution for quantizing SSM inputs. A more comprehensive study on quantizing SSMs using floating-point quantization is left for future work.

## L  LOW BIT-WIDTH QUANTIZATION

We show that Transformer-based quantization methods do not generalize well to Mamba blocks. We re-implement state-of-the-art low bit-width quantization methods for Transformers, Quip# (weight-only quantization, W2A16) (Tseng et al., 2024) and QuaRot (W4A4) (Ashkboos et al., 2024b), for the Mamba structure. These are denoted by Mamba-Quip# and Mamba-Had, respectively Our experiments show that they fail to effectively quantize Mamba in low bit-width settings, as shown in Tables 12 and 13. In Table 12, applying Quip# and QuaRot to Mamba results in much higher perplexity, leading to worse performance compared to Transformers. For instance, Mamba-Quip# quantizes Llama-2-7b with W2A16, causing only a $1.02\times$ increase in perplexity, whereas our implementation on Mamba results in a $1.34\times$ increase. In Table 13, although models with different bit-widths cannot be directly compared, the results show that both Mamba-Quip# and Mamba-Had reduce the average accuracy across six zero-shot downstream tasks. Both Mamba-Had and Quamba fail at the 4-bit level, reducing the average accuracy from 63.1% to 30.2% and 29.3%, respectively. Our study highlights that quantizing SSMs is particularly challenging, especially in low bit-width settings, as their input and output activations exhibit a causal relationship with varying levels of outliers, a phenomenon unique to SSMs (*ref.* Section M). We note that similar results were found recently for large transformer-based LLMs which were shown to have best performance under memory constraints for a bit-width of 6-8, with worse results for a bit-width of 4 (Kumar et al., 2024).

Table 12: Quantizing Llama-2-7b and Mamba 2.8B with low bit-width methods. The perplexity on Wiki2 is reported.

| Methods | Precision | Llama-2-7b | Mamba-2.8b |
|---|---|---|---|
| Mamba | FP16 | 5.47 | 9.45 |
| Mamba-Quip# | W2A16 | 5.56 (1.02 $\times$) | 12.71 (1.34 $\times$) |
| Mamba-Had | W4A4 | 6.10 (1.11 $\times$) | failed |
| Quamba (Ours) | W8A8 | - | **9.91** |

Table 13: Quantizing Mamba 2.8B with low bit-width methods. The average accuracy on six zero-shot tasks is reported.

| Methods | Precision | Mamba-2.8b |
|---|---|---|
| Mamba | FP16 | 63.1% |
| Mamba-Quip# | W2A16 | 58.5% |
| Mamba-Had | W4A4 | 30.2% |
| Quamba (Ours) | W4A4 | 29.3% |
| Quamba (Ours) | W8A8 | **62.2%** |

## M  COMPARING SSMS WITH SELF-ATTENTION LAYERS

**Comparing the quantization sensitivity.** We explore the quantization sensitivity of input and output activation maps for both SSMs and self-attention layers, as shown in Figure 11. We conduct experiments on Mamba 2.8B, a same-sized Transformer Pythia 2.8B, and a recently published, comparably sized Transformer Llama 3.2 3B. Quantizing the input $x$ and output $y$ tensors leads to the most significant accuracy drop on the LAMBADA dataset for SSMs, compared to other tensors. In contrast, quantizing the tensors in self-attention layers (*i.e.* $h$, $q$, $k$, $v$, and output $y$) results in minimal accuracy loss. For transformers, using 8-bit $h_d$ tensors in feedforward layers significantly degrades model performance. Due to the differing quantization sensitivity patterns between Mamba and Transformer blocks, we highlight the *smooth*, *outlier*, and *sensitive* paths for both in Figure 12. This highlights that different quantization methods are needed for SSM-based models.

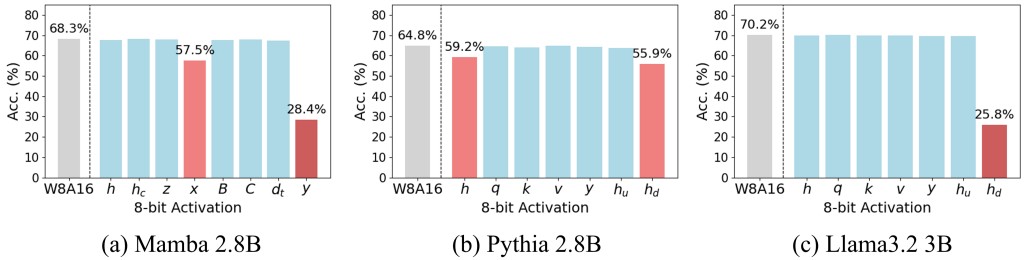

(a) Mamba 2.8B      (b) Pythia 2.8B      (c) Llama3.2 3B

Figure 11: Quantizing the input $x$ and output $y$ tensors causes the most significant accuracy drop on the LAMBADA dataset for SSMs. In contrast, quantizing the tensors in self-attention layers results in minimal accuracy loss.

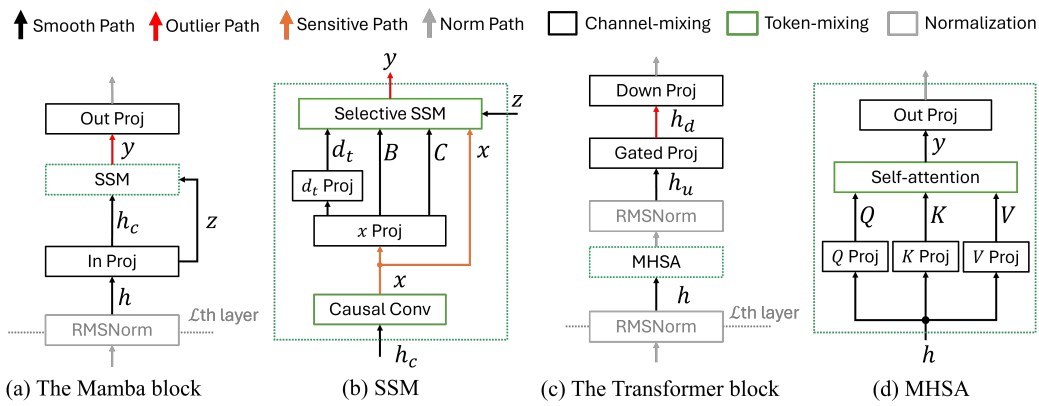

(a) The Mamba block      (b) SSM      (c) The Transformer block      (d) MHSA

Figure 12: We annotate the different sensitivity patterns for SSMs and self-attention layers, which shows that they require different quantization techniques. The residual connections are simplified to avoid cluttering the figure.

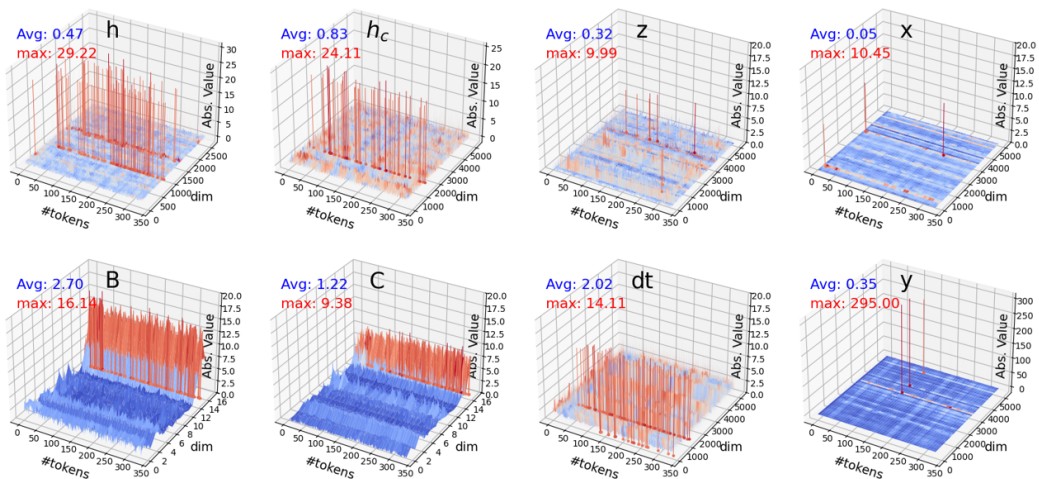

Figure 13: We visualize the activations of the SSM in the last layer of Mamba 2.8B. This figure shows that the values in the $x$ tensor are not significantly large. However, outliers are present in the $y$ tensor of the SSM output, making them difficult to represent in low bit-width data types.

**Visualizing activations.** We visualize these activation maps in Figures 13 and 14. Figure 13 shows the SSM activation maps in the last layer of Mamba 2.8B, while Figure 14 shows those for the self-attention layer in Llama 3.2 3B. We use the same test sample to visualize these activations.

Notably, SSMs and self-attention layers exhibit *distinct* activation patterns. Outliers appear in the SSM output $y$, unlike in self-attention layers, where the outputs $y$ remain smooth and *do not* present outliers. In Transformer blocks, outliers only occur in the $h_d$ of the feedforward layers, making them difficult to quantize. In contrast, Mamba blocks have large outliers in the SSM output $y$ tensor, which are challenging to represent with low-bit data types. While the $x$ tensor values are not significantly large, they are highly sensitive to quantization errors. We address this issue in Mamba blocks and introduce Quamba to manage the unique quantization patterns of SSMs.

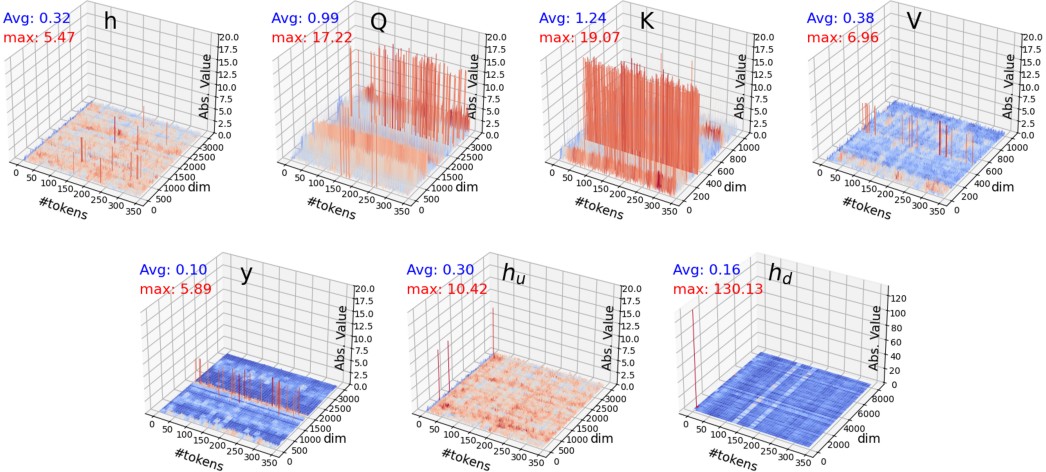

Figure 14: We visualize the activations of the self-attention layer in the last layer of Llama 3.2 3B. This figure shows that outliers are *not* present in the output $y$ of the self-attention layers. In contrast to SSMs, the outlier issue arises in the $h_d$ tensor of the *feedforward layers*.

**Comparing the precision mapping.** Figure 15 compares our precision mapping to the standard one used in Transformers (Zhao et al., 2023; Ashkboos et al., 2024b; Lin et al., 2024; Xiao et al., 2023).

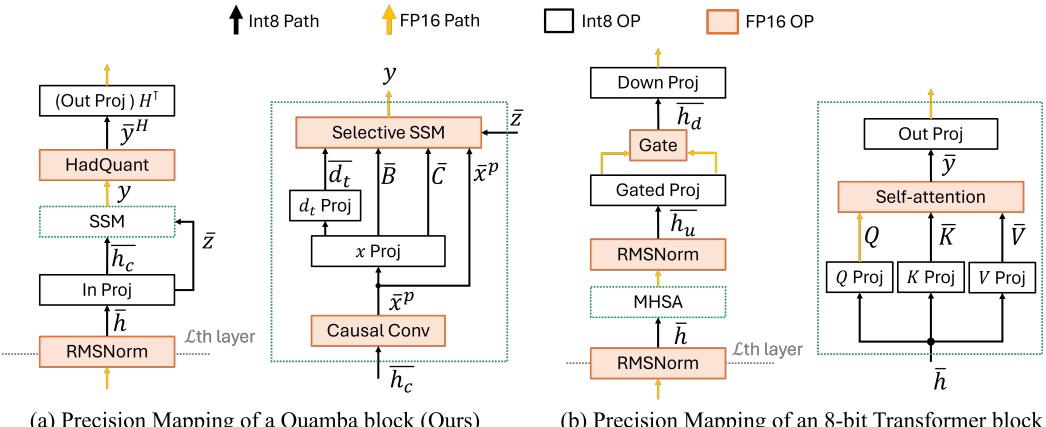

(a) Precision Mapping of a Quamba block (Ours)  (b) Precision Mapping of an 8-bit Transformer block

Figure 15: We compare our precision mapping to the precision mapping commonly used in Transformers. The residual connections are simplified to avoid cluttering the figure.

## N    QUANTIZATION ERROR ANALYSIS FOR SSMS

### N.1    EMPIRICAL ANALYSIS OF A MATRICES

Given a discrete linear time-invariant (LTI) state space model: $h(t) = A(t)h(t-1) + Bx(t)$, we unroll the equation at time step $t$:

$$h(t) = A(t) \cdots A(2)Bx(1) + A(t) \cdots A(3)Bx(2)$$
$$+ \cdots + A(t)A(t-1)Bx(t-2) + A(t)Bx(t-1) + Bx(t).$$

We visualize the spectral norm of the cumulative production of A matrices (*i.e.,* $||A(t)A(t-1)\cdots||_2$) from pre-trained Mamba-130m in Figure 16. The figure shows that SSM puts more weight on recent history by performing cumulative production of the A matrix. We leverage this property from SSMs to derive our quantization error bound and show the quantization error is bounded with respect to the input sequence length $T$.

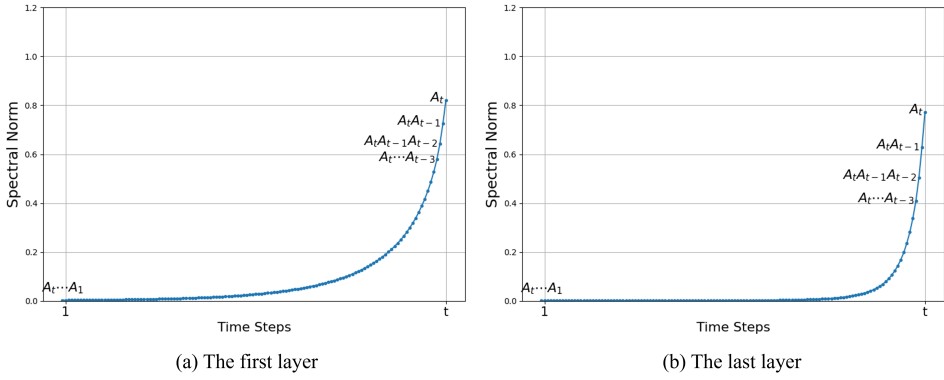

(a) The first layer                                  (b) The last layer

Figure 16: The spectral norm of the cumulative production of A matrices from pre-trained Mamba-130m, *i.e.,* $||A(t)A(t-1)\cdots||_2$.

### N.2    THEORETICAL ERROR ANALYSIS

We consider a discrete linear time-invariant (LTI) state space model at time step $t$ defined as $h(t) = A(t)h(t-1) + Bx(t)$, where the state matrix $A(t) \in \mathbf{R}^{N \times N}$, input matrix $B \in \mathbf{R}^{N \times P}$, implicit latent state $h(t) \in \mathbf{R}^{N \times 1}$, and input $x(t) \in \mathbf{R}^{P \times 1}$. Based on the behavior featured in Figure 16, we assume the norm of the state matrix A can be bounded by an exponential function such that $||A(t)||_2 \leq a \cdot e^{t-T}$, and the input matrix $B$ is bounded by $||B|| \leq b$, where $0 < a < 1$, $b > 0$, and $1 \leq t \leq T$. In our proof, we utilize the spectral norm ($||\cdot|| = ||\cdot||_2$). Furthermore, we assume the system input contains a quantization error $\delta_x(t) \in \mathbf{R}^{P \times 1}$, such that $\overline{x}(t) = x(t) + \delta_x(t)$, where $||\delta_x(t)||_2 \leq \epsilon$ and $\epsilon > 0$. The system is initialized as $h(0) = [0, 0, \ldots 0]^T$.

**Theorem N.1.** *The quantization error of $\Delta(t) = \overline{h}(t) - h(t)$ at time step $t$ for the given discrete linear time-invariant model is bounded such that: $||\Delta(t)||_2 \leq \epsilon b\left(\frac{1}{1-ae^{t-T}}\right)$. Consequently, the* **global** *quantization error (*i.e.,* $t = T$) is bounded by : $||\Delta(T)||_2 \leq \frac{\epsilon b}{1-a}$.*

*Proof.* Given the quantization error $||\delta_x(t)||_2 \leq \epsilon$ and the quantized input $\overline{x}(t) = x(t) + \delta_x(t)$ of each step, we have an original system $h(t)$ and a quantized system $\overline{h}(t)$:

$$h(t) = A(t)h(t-1) + Bx(t)$$
$$\overline{h}(t) = A(t)\overline{h}(t-1) + B(x(t) + \delta_x(t)).$$

By subtracting two systems, we have quantization error $\Delta(t) = \overline{h}(t) - h(t)$ given by: $\Delta(t) = A(t)(\overline{h}(t-1) - h(t-1)) + B\delta_x(t) = A(t) \cdot \Delta(t-1) + B\delta_x(t)$. Let's consider the recurrence on

quantization error, assuming that $\Delta(0) = 0$:

$$t=1, \Delta(1) = A(1)\Delta(0) + B\delta_x(1) = B\delta_x(1)$$
$$t=2, \Delta(2) = A(2)\Delta(1) + B\delta_x(2) = A(2)B\delta_x(1) + B\delta_x(2)$$
$$t=3, \Delta(3) = A(3)\Delta(2) + B\delta_x(3) = A(3)A(2)B\delta_x(1) + A(3)B\delta_x(2) + B\delta_x(3)$$
$$t=4, \Delta(4) = A(4)\Delta(3) + B\delta_x(4) = A(4)A(3)A(2)B\delta_x(1) + A(4)A(3)B\delta_x(2)$$
$$+ A(4)B\delta_x(3) + B\delta_x(4)$$

$$\cdots$$

At time step $t$, we have

$$
\begin{aligned}
\Delta(t) &= A(t)\Delta(t-1) + B\delta_x(t) \\
&= \Pi_{u=2}^t A(u)B\delta_x(1) + \Pi_{u=3}^t A(u)B\delta_x(2) + \Pi_{u=4}^t A(u)B\delta_x(3) \\
&\quad + ... + A(t)B\delta_x(t-1) + B\delta_x(t) \\
&= \Sigma_{v=2}^t \big( \Pi_{u=v}^t A(u)B\delta_x(v-1) \big) + B\delta_x(t).
\end{aligned}
\tag{4}
$$

Let's consider quantization error $\Delta(t)$ such that $||\Delta(t)||_2 = ||\overline{h}(t) - h(t)||_2$:

$$
\begin{aligned}
||\Delta(t)||_2 &= ||\Sigma_{v=2}^t \big( \Pi_{u=v}^t A(u)B\delta_x(v-1) \big) + B\delta_x(t)|| \\
&\leq ||\Sigma_{v=2}^t \big( \Pi_{u=v}^t A(u)B\delta_x(v-1) \big)|| + ||B\delta_x(t)|| \\
&\leq ||\Sigma_{v=2}^t \big( \Pi_{u=v}^t A(u)B\delta_x(v-1) \big)|| + \epsilon b \\
&\leq \epsilon b ||\Sigma_{v=2}^t \big( \Pi_{u=v}^t A(u) \big)|| + \epsilon b \\
&\leq \epsilon b \Sigma_{v=2}^t ||\Pi_{u=v}^t A(u)|| + \epsilon b \\
&\leq \epsilon b \Sigma_{v=2}^t \Pi_{u=v}^t ||A(u)|| + \epsilon b \\
&\leq \epsilon b \Sigma_{v=2}^t \Pi_{u=v}^t a e^{u-T} + \epsilon b \\
&\leq \epsilon b (\Sigma_{v=2}^t a^{t-v+1} \Pi_{u=v}^t e^{v-T} + 1) \\
&= \epsilon b (\Sigma_{i=1}^{t-1} a^i \Pi_{j=0}^{i-1} e^{t-j-T} + 1) \\
&= \epsilon b (\Sigma_{i=1}^{t-1} a^i e^{(t-T)\cdot i} \Pi_{j=0}^{i-1} e^{-j} + 1) \\
&\leq \epsilon b (\Sigma_{i=1}^{t-1} \Big( \frac{ae^t}{e^T} \Big)^i + 1) \\
&= \epsilon b (\frac{1 - (\frac{ae^t}{e^T})^t}{1 - (\frac{ae^t}{e^T})}) \\
&\leq \epsilon b (\frac{1}{1 - ae^{t-T}})
\end{aligned}
$$

The time-step dependent bound becomes: $||\Delta(t)||_2 \leq \epsilon b (\frac{1}{1-ae^{t-T}})$. Therefore, the global bound for a given sequence length T (*i.e.,* $t = T$) is $||\Delta(T)||_2 \leq \frac{\epsilon b}{1-a}$. $\qquad \square$

## O  LIMITATIONS AND BROADER IMPACTS

We find that the accuracy degradation is not negligible in both accuracy and perplexity in Table 7 and Table 2. Despite this, the performance trade-off is acceptable given the significant improvements in latency and resource efficiency. Our work enables large language models to be deployed on resource-limited devices. As a positive feature, our method may push the development of privacy-centric on-device applications, where sensitive data can be processed locally without relying on cloud services. However, our work may also present challenges such as increased device resource consumption and potential security vulnerabilities if the local devices are compromised.

