# OpenReview forum: "Quamba: A Post-Training Quantization Recipe for Selective State Space Models"
_ICLR.cc/2025/Conference — ICLR 2025 Poster_

### Official Review · Reviewer_8GLX · 2024-10-23

**Soundness:** 3
**Presentation:** 3
**Contribution:** 2
**Rating:** 6
**Confidence:** 4

**Summary:**

This paper proposes a post-training quantization method for state space models (SSMs), which enables the utilization of low-bit acceleration in modern GPUs and achieves notably lower generation latency and small accuracy penalty. The method is mainly comprised of two techniques: 1) static 8-bit per-tensor quantization for input activations, and 2) Hadamard transform for outlier-free output activation quantization.

**Strengths:**

- Overall, the paper is well-structured, well-motivated, and well-written. The right amount of background introduction, motivating examples, and explanation of challenges make this paper easy to understand.
- The paper did thorough experiments to show the generalizability and scalability of the proposed method on various SSM-based implementations. In the results, the proposed Quamba shows a clear latency improvement and makes it possible to run large-scale models on devices with a limited amount of memory.

**Weaknesses:**

- The novelty of the proposed method is not clear. As mentioned in the text, the paper acknowledges that clipping outliers in input activations with a percentile max is a widely applied technique, and smoothing out the outliers with the Hadamard transform is also not new and has been applied in prior quantization methods for transformer attention computation. The main novelty of this paper therefore may come from being the first to apply these two techniques to SSM.
- There is no significant benefit over QuaRot-SSM, which is a re-implementation of QuaRot for Mamba. The positive side is that this does serve as a strong baseline, however, their performance is very similar in many experiments. Figure 7 in the appendix, to some extent, also shows that the differences in architecture are small.

**Questions:**

- Could the paper articulate more clearly what aspects of applying the two proposed techniques (clipping and Hadamard transform) to SSMs are novel or challenging compared to their use in other contexts?
- Could the paper provide a better positioning of this paper, in terms of what the main difference is between the proposed method and the prior QuaRot? It seems that a trivial change on the original QuaRot (such as removing the extra Hadamard transform, which is a natural choice) can easily improve the performance.
- In Figure 1 (b) and (c), could the paper provide the data for QuaRot-SSM for a more complete comparison?
- In Figure 2 (b) and (c), it is unclear what y_t means. And why are there multiple (colors of) MSE curves? For example, for the curve of V_bar (red one in (b)), does it still represent the MSE of y_t and how should we understand this curve? Figure 3 (a) is much clearer in this sense.
- Is it possible to support 4-bit quantization with the proposed method?

---

> ### Author Response · Authors · 2024-11-21
>
> Thank you for your insightful review of our submission. We have outlined responses to your concerns and questions below.
>
> > The novelty of the proposed method is not clear.  Could the paper articulate more clearly what aspects of applying the two proposed techniques (clipping and Hadamard transform) to SSMs are novel or challenging compared to their use in other contexts?
>
> We apply clipping and Hadamard transform **because they are effective in terms of performance and latency** with suitable non-trivial changes.  Quamba achieves a Pareto-optimal balance between accuracy and latency compared to Transformer baselines, **highlighting its uniqueness and novelty in SSM quantization** (see Figure 7 in Appendix C). As shown in Table R1 and the updated Figure 1 (b), the re-implemented Mamba-Had (W8A8) **does not offer latency benefits** over the FP16 model on A5000 . In contrast, **Quamba offers 1.7× and 1.2× speedups** on edge and cloud GPUs, significantly outperforming Mamba-Had.
>
> **We set out to address what other papers have not: State Space Model (SSM) quantization**. SSM quantization is very challenging due to a key finding in our paper: *Quantizing SSM input leads to large deviations in the output due to the linear recurrence, and SSM output has large outliers*. This finding is **unique** to SSMs and different from self-attention layers in Transformers, as shown in Figure 2. We also show that most prior techniques, including the “highly effective self-attention techniques”:  such as SmoothQuant [12] and QuaRot [13], **fail** at quantizing the SSM with acceptable accuracy or offering latency improvements on cloud or edge devices.
>
> Furthermore, off-the-shelf quantization methods fail for Mamba blocks in Cobra [11] and the official Jamba on Hugging Face [9, 10] , thereby prompting the need to keep them in FP16. **Quamba fills the missing puzzle in this space with its effectiveness and efficiency for SSM quantization**. We also validate Quamba on Jamba, the largest SSM-Transformer hybrid model, as shown in Table 4.
>
> We summarize our novelty and contributions:
> - **We explicitly show that SSM quantization is non-trivially challenging and current methods designed for Transformers do not generalize well to SSMs**
> - **We present an in-depth analysis of the causal relationship between input and output activations in SSMs and Transformers**
> - **To address these challenges, we employ the clipping and Hadamard transform and propose Quamba, a solution tailored for SSMs, supported by theoretical proof and validated speedup**
> - **Our work fills a critical gap in the current research on quantization for large-scale Mamba-Transformer hybrid LLMs such as Jamba**
>
> > There is no significant benefit over QuaRot-SSM, which is a re-implementation of QuaRot for Mamba.
>
> Quamba outperforms both the original transformer-based quantization QuaRot [13] and our re-implemented version on the Mamba backbone which we dub **Mamba-Had** (QuaRot-SSM in the original version). **Quamba offers 1.7× and 1.2× speedups** on edge and cloud GPUs, significantly outperforming Mamba-Had. In contrast, the re-implemented QuaRot for the Mamba backbone Mamba-Had (W8A8) **does not offer latency benefits** over the FP16 model on A5000, as shown in Table R2 and the updated Figure 1 (b).
>
> We report seconds (sec.) for all time-to-last-token (TTLT) latencies on A5000 and the average accuracy among six zero-shot tasks in Table R2.
>
> | Table R2             | Bit-width  |  Latency (sec.)   |Avg. Acc.  |
> |-----------------------|:--:|-----------------------|-----------|
> |                            |  | 1k &nbsp; / 2k &nbsp; / 4k &nbsp; &nbsp; /  8k    |        |
> | Mamba-2.8B      | FP16 | 5.1 / 10.3 / 20.7 / 41.5  |  63.1%    |
> | Mamba-Had       | W8A8 | 5.5 / 10.8 / 21.9 / 43.8  |  62.4%    |
> | Quamba-2.8B     | W8A8  | **4.1** / **8.4** &nbsp; / **16.9** / **33.8**  |  62.2%    |
> | Quamba-2.8B-O  | W8A8  | **4.1** / **8.4** &nbsp; / **16.9** / **33.8**  |  **62.5%**    |
>
> In response to the reviewers’ suggestions, we applied a clipping optimization algorithm [5] to Quamba, naming the enhanced version Quamba-O. This modification improves our average accuracy and **outperforms Mamba-Had in both accuracy and latency**. We highlight our innovations in proposing an **efficient quantization framework for SSMs** that achieves Pareto optimality on both edge and cloud GPUs. Optimizing clipping range is complementary to our key contributions.

---

> ### Author Response · Authors · 2024-11-21
>
> > In Figure 1 (b) and (c), could the paper provide the data for QuaRot-SSM for a more complete comparison?
>
> We rename QuaRot-SSM to Mamba-Had since it is not QuaRot [13] but rather the Mamba architecture with the Hadamard transform previously introduced for transformer LLMs in [13].
>
> We have updated Figure 1(b) for comparison based on your comment and extract the updated numbers in Table R2. As shown in Table R2 and the updated Figure 1 (b), the re-implemented Mamba-Had (W8A8, QuaRot-SSM in our original version) does not offer latency benefits over the FP16 model on A5000.
>
> Due to the nature of SSMs, which compress previous history with a constant memory size (i.e., no K-V cache), the memory footprint for both Quamba and Mamba-Had will remain the same.
>
>
> > In Figure 2 (b) and (c), it is unclear what $y_t$ means. And why are there multiple (colors of) MSE curves?
>
> $y_t$ denotes the output from the SSM or the self-attention at each time step $t$. The multiple colors are used to show that we quantize **different input tensors** and **measure the normalized MSE at the output** for each timestep t.
>
> We conduct a sensitivity analysis on quantizing each input to the SSM and self-attention layers. More specifically, we **quantize one input tensor at a time** and **measure the normalized MSE at the output** at each time step t (i.e., $y_t$). Thus, for the self-attention layer, there are four lines representing input with quantized $Q$, $K$, $V$, and $h$, while for the SSM, the four lines represent input with quantized $B$, $C$, $\Delta$, and $x$. For example, Figure 2(b) shows that SSMs are highly sensitive to quantization errors in the input $x$, resulting in significant deviations in the output $y$, which is **unique** to SSMs. In contrast, quantizing $B$, $C$ and $\Delta$ does not introduce substantial errors in the output $y$. For comparison, in Figure 2(c), self-attention layers are more resilient to quantization errors in the input h, despite $Q$, $K$, and $V$ being $h$-dependent.
>
> Our technique addresses this specific challenge in SSMs by enhancing quantization precision for the $x$ tensor in a hardware-friendly way, such as through clipping, which significantly reduces errors in the output y while lowering latency on hardware platforms.
>
>
> > Is it possible to support 4-bit quantization with the proposed method?
>
> We have evaluated both Quamba and Mamba-Had under the W4A4 setting.  However, as shown in Table R4, **both** Mamba-Had (QuaRot [13] re-implemented for Mamba) and Quamba **fail** at the 4-bit level, reducing the average accuracy from 63.1% to 30.2% and 29.3%, respectively. This **highlights the challenges SSMs present in model quantization** under the low bit-width setting. We note that similar results were found recently for large transformer-based LLMs which were shown to have **best performance under memory constraints for a bit-width of 6-8, with worse results for a bit-width of 4 [14]**.
>
> | Table R4        | Bit-width | Avg. Acc. |
> |-----------------|-----------|-----------|
> | Mamba-2.8B      | FP16      | 63.1%     |
> | Mamba-Had-2.8B  | W4A4      | 30.2%     |
> | Quamba-2.8B     | W4A4      | 29.3%     |
>
> As a result, our work focuses on SSM quantization under the **W8A8** setting. In this case, **Quamba clearly outperforms Mamba-Had by achieving 1.7× and 1.2× speedups** on edge and cloud GPUs. In contrast, Mamba-Had (W8A8) fails to provide latency benefits over the FP16 model on the A5000, as shown in Table R2 and the updated Figure 1(b). We report seconds (s) for all time-to-last-token (TTLT) latencies on A5000 and the average accuracy among six zero-shot tasks in Table R2.
>
> | Table R2             | Bit-width  |  Latency (sec.)   |Avg. Acc.  |
> |-----------------------|:--:|-----------------------|-----------|
> |                            |  | 1k &nbsp; / 2k &nbsp; / 4k &nbsp; &nbsp; /  8k    |        |
> | Mamba-2.8B      | FP16 | 5.1 / 10.3 / 20.7 / 41.5  |  63.1%    |
> | Mamba-Had       | W8A8 | 5.5 / 10.8 / 21.9 / 43.8  |  62.4%    |
> | Quamba-2.8B     | W8A8  | **4.1** / **8.4** &nbsp; / **16.9** / **33.8**  |  62.2%    |
> | Quamba-2.8B-O  | W8A8  | **4.1** / **8.4** &nbsp; / **16.9** / **33.8**  |  **62.5%**    |

---

> > ### Comment · Reviewer_8GLX · 2024-11-22
> >
> > I would like to thank the authors for their very detailed responses and extra experiment results.
> >
> > In the caption of Figure 8 (in the revised manuscript), the paper states that QuaRot requires extra Hadamard transforms and transpositions for SSM input activations. For this statement, I have some follow-up questions:
> >
> > - Is this the major difference between Mamba-Had and Quamba?
> > - Why does Quamba remove the Hadamard transforms on input activations? What is the motivation behind it? The paper should convince the reader that this is not a trivial modification and that it is not just an unnecessary overhead for Mamba-Had.
> > - Does the latency improvement mainly come from this change?
> >
> > These questions should be clarified in the revised manuscript and some of the discussion about Figure 8 should be in the main text rather than the appendix.

---

> ### Author Response · Authors · 2024-11-22
>
> We greatly appreciate the time and effort you put into reviewing our submission. We address the follow-up questions below.
>
> > Is this the major difference between Mamba-Had and Quamba?
>
> The key difference is that we apply clipping to enhance the quantization precision of the x tensor, demonstrating it to be a **more efficient strategy** specifically for SSMs. Because causal convolution and selective scan are unique to SSMs, we find that simply applying the Hadamard transform, as adapted from Transformers, is particularly **inefficient** (see the answers below for more details).
>
> We propose Quamba based on several **key observations** on **SSMs' properties** and **architectures**. Our approach **minimizes** the need for matrix transpositions, memory reorganization, and online Hadamard transformations, resulting in speedups during both the generation and decoding stages.  (see below answers for more details)
>
> > Does the latency improvement mainly come from this change?
>
> **Yes**, we observed that the forward and backward Hadamard matrices **cannot be fused** within the causal convolution (forward transform) and selective scan (backward transform), leading to significant computational overhead.
>
> **The additional online Hadamard transforms are particularly costly** because they require extra matrix transpositions and memory contiguous due to mismatched output and input shapes between the causal convolution and selective scan. Specifically, the process involves the following sequence:
>
> Causal Conv -> **[b, dim, seqlen]** -> transpose & contiguous  -> [b, seqlen, dim]  -> forward Hadamard & quantize -> [b, seqlen, dim] -> backward Hadamard -> [b, seqlen, dim] -> transpose & contiguous -> **[b, dim, seqlen]** -> selective scan.
>
> In contrast, our method avoids these additional transpositions and memory contiguous steps. The streamlined process is as follows:
>
> Causal Conv & quantize -> **[b, dim, seqlen]** -> selective scan.
>
> For clarity and to avoid overloading Figure 8, we have omitted the transposition and memory contiguous steps in the illustration.
>
> In summary, our method significantly reduces the reliance on matrix transpositions, memory reorganization, and online Hadamard transforms, delivering **speedups during both the generation and decoding stages**.
>
>
>
> > Why does Quamba remove the Hadamard transforms on input activations? What is the motivation behind it? The paper should convince the reader that this is not a trivial modification and that it is not just an unnecessary overhead for Mamba-Had.
>
> We remove the Hadamard transforms applied to the input activations based on key observations and motivations outlined in our paper.
>
> ### Finding:
> - The SSM is highly sensitive to quantization errors in the x tensor, as illustrated in Figure 2(c). Therefore, a primary goal of our work is to enhance the quantization precision of the x tensor.
>
> ### Observations:
> - The Hadamard matrices **cannot be fused** in the causal convolution (forward transform) and selective scan (backward transform), resulting in significant computational overhead, as shown in Figure 8 (b) and the above answer.
> - The numerical values in the x tensor are relatively **small (typically <10)**, as discussed in lines 270–279 and shown in Figure 3(a) and the left panel of Figure 9.
>
> These observations motivate us to **explore a more efficient algorithm** for quantizing the x tensor without relying on the Hadamard transform.
>
> We find that applying clipping to enhance the quantization precision of the x tensor proves to be a more efficient strategy. This approach **places Quamba ahead of Mamba-Had** on the Pareto front. Additional quantization variants for the  x tensor are provided in Appendix G, Table 9.
>
>
> > These questions should be clarified in the revised manuscript and some of the discussion about Figure 8 should be in the main text rather than the appendix.
>
> We have incorporated your suggestion by adding the discussion at lines 390–391 and have also highlighted the relevant discussion in our original text. Thank you for your valuable input.

---

> > ### Comment · Reviewer_8GLX · 2024-11-23
> >
> > The authors' response answered all my questions and the revision has significantly improved the quality of this manuscript. Therefore, I will raise my rating accordingly. Thanks again for the detailed and insightful response from the authors.

---

> > > ### Author Response · Authors · 2024-11-23
> > >
> > > Thank you for your thoughtful and careful review of our paper. We truly appreciate your positive feedback and your rating, which indicates a leaning toward accepting our submission.

---

### Official Review · Reviewer_evxv · 2024-10-28

**Soundness:** 2
**Presentation:** 1
**Contribution:** 3
**Rating:** 6
**Confidence:** 5

**Summary:**

The paper studies quantization of stat-space models. The propose Quamba which employs 8-bit quantization and Walsh-Hadamard transformations of activations to preserve accuracy.

**Strengths:**

- Studies quantization of SSMs, which is novel since SMMs themselves are new and quantization studies have not caught up with them.
- The empirical results are not just scoped to accuracy studies using fake quantization, but actual profiling is done using INT8 and cutlass.

**Weaknesses:**

- Empirically, Quamba does not show advantages compared to Quarot.
- MSE studies are used to compare difficulty of quantizing attention vs SSM (e.g., Fig 2). However, absolute numbers for MSE are not useful when inputs and models are different. A better way to visualize the quantization difficulty would be to employ a normalized MSE. In other words, the L2 norm of the delta divided by the L2 norm of the unquantized tensor, rather than just the L2 norm of the delta.
- Theorem 4.1 is very badly written, I had to refer to the appendix to guess what the authors meant. Note that lowercase b and epsilon were never defined and they are the defining variables of the theorem! Please fix this and introduce b as being an element from the B matrix, and epsilon to be an upper bound of the quantization step. P.S., these are my guesses for what was meant.
-While I appreciate Theorem 4.1 which states that quantization error compounds over time-steps (correct me if I am wrong). Where was this used? Which aspect of the proposed Quamba leverages Theorem 4.1? It seems like this was just added as decorative math.

**Questions:**

- Why is TTLT used? It makes more sense to evaluate latency (TTFT) and throughput (token-to-token).
- Since extremely large outliers are identified as problematic for SSMs, have the authors considered using clipping to lower quantization noise. For example, OCTAV may be useful in this setup [1]
- Empirical results with comparisons to transformers are focused on the W8A8 case. However, Transformers have been quantized to 4-bit (e.g., using Quarot).Is it therefore a fair comparison to use 8-bit transformers?

[1] Sakr, Charbel, et al. "Optimal clipping and magnitude-aware differentiation for improved quantization-aware training." International Conference on Machine Learning. PMLR, 2022.

----
Good responses, I am raising to 6.

---

> ### Author Response · Authors · 2024-11-21
>
> We sincerely value your comprehensive review of our submission and address your concerns and questions as outlined below.
>
> > Quamba does not show advantages compared to Quarot.
>
> We note that **QuaRot-SSM in our paper is not QuaRot [13]** but rather the Mamba architecture with the Hadamard transform previously introduced for transformer LLMs in [13]. We re-implement a version on the Mamba backbone which we dub **Mamba-Had** (QuaRot-SSM in the original version).
>
> Quamba outperforms both the original transformer-based quantization QuaRot [13] and Mamba-Had. **Quamba offers 1.7× and 1.2× speedup on edge and cloud GPUs, significantly outperforming Mamba-Had**. In contrast, the re-implemented QuaRot for the Mamba backbone Mamba-Had (W8A8) **does not offer latency benefits** over the FP16 model on A5000, as shown in Table R2 and the updated Figure 1 (b).  We apply clipping and Hadamard transform **because they are effective in terms of performance and latency** with suitable non-trivial changes. To support our claims, we show that Quamba achieves a Pareto-optimal balance between accuracy and latency compared to Transformer baselines, highlighting its uniqueness and novelty in SSM quantization (see Figures 7 in Appendix C).
>
> We report seconds (sec.) for all time-to-last-token (TTLT) latencies on A5000 and the average accuracy among six zero-shot tasks in Table R2.
>
> | Table R2             | Bit-width  |  Latency (sec.)   |Avg. Acc.  |
> |-----------------------|:--:|-----------------------|-----------|
> |                            |  | 1k &nbsp; / 2k &nbsp; / 4k &nbsp; &nbsp; /  8k    |        |
> | Mamba-2.8B      | FP16 | 5.1 / 10.3 / 20.7 / 41.5  |  63.1%    |
> | Mamba-Had       | W8A8 | 5.5 / 10.8 / 21.9 / 43.8  |  62.4%    |
> | Quamba-2.8B     | W8A8  | **4.1** / **8.4** &nbsp; / **16.9** / **33.8**  |  62.2%    |
> | Quamba-2.8B-O  | W8A8  | **4.1** / **8.4** &nbsp; / **16.9** / **33.8**  |  **62.5%**    |
>
> > MSE studies are used to compare difficulty of quantizing attention vs SSM (e.g., Fig 2).  A better way to visualize the quantization difficulty would be to employ a normalized MSE.
>
> Upon reviewing Fig. 2’s implementation, we discovered our y-axis label is incorrect. Our figure is actually already reporting “normalized MSE”, so it already is shown as the reviewer would like it presented. We thank the reviewer for raising this point.
>
> > Why is TTLT used? It makes more sense to evaluate latency (TTFT) and throughput (token-to-token).
>
> TTLT represents the latency to full response, as user experience is impacted by both when the response starts and when the entire response is fully generated, as shown in Figure 1(b). **Our method provides speedup at all stages** and therefore we report improvements in  all three: TTFT (time-to-first-token), TPOT (time-per-output-token, i.e, token-to-token), and TTLT (time-to-last-token), as outlined in lines 374, 375, and 395.
>
> To alleviate the reviewer’s concerns, we extract the TTFT profiling from Table 1 to Table R7. Additionally, we use the official implementation from QuaRot to profile TTFT for Llama-2-7B (W4A4KV4) on A5000 for reference purposes. **The quantized W4A4KV4 Llama-2-7B is roughly comparable in model size to Quamba (8-bit Mamba)**, making it a reasonably equivalent basis for comparison. We report milliseconds (msec.) for all latencies on A5000 in Table R7.
>
> | Table R7	               |  Bit-width  | Model size  |   Latency (msec.)	 |Avg. Acc.  |
> |-------------------------------|----------------|:------:|----------------|----------|
> |                                      |                  |       |  512 &nbsp;&nbsp; &nbsp;&nbsp;/ 1024 &nbsp;&nbsp;  / 2048    |             |
> | Llama-2-7B (QuaRot)  | W4A4KV4 | 3.5G | 155.94 / 234.75 / 384.62 | 65.6%   |
> | Mamba-2.8B               |  FP16   | 5.3G | 61.19&nbsp; &nbsp;/ 102.29 / 184.16 | 63.1%   |
> | Mamba-Had-2.8B      |  W8A8   | 2.7G | 56.87&nbsp;&nbsp; / 103.98 / 199.83 | 62.4%   |
> | Quamba-2.8B            |  W8A8   | 2.7G | **48.24** &nbsp;&nbsp;/ **84.84** &nbsp; / **165.13** | 62.2%   |
> | Quamba-2.8B            |  W8A8   | 2.7G | **48.24** &nbsp;&nbsp;/ **84.84** &nbsp; / **165.13** | **62.5%**   |

---

> ### Author Response · Authors · 2024-11-21
>
> > Have the authors considered using clipping to lower quantization noise? For example, OCTAV may be useful in this setup [5]
>
> We thank the reviewer for suggesting this reference. We have implemented the method [5] you suggested, as shown in Table R8.  We report the average accuracy among six zero-shot tasks and seconds (sec.) for all time-to-last-token (TTLT) latencies on A5000 in Table R8.
>
> |     Table R8  | Bit-width| 130m  | 370m  | 1.4B  | 2.8B  | 1K latency (sec.) |
> |------------|:--:|-------|-------|-------|-------|-----------|
> | Mamba       | FP16 | 44.7% | 51.7% | 59.7% | 63.1% |    5.1    |
> | Mamba-Had   | W8A8 | 42.6% | 50.0% | 58.1% | 62.4% |    5.5    |
> | Quamba     | W8A8  | **43.5%** | 49.0% | 58.4% | 62.2% |    **4.1**    |
> | Quamba-O   | W8A8  | 42.9% | **50.4%** | **58.4%** | **62.5%** |    **4.1**    |
>
>
> We would like to emphasize that our innovations lie in proposing an **efficient quantization framework for SSMs** that achieves Pareto optimality on both edge and cloud GPUs. Indeed, clipping range optimization [5] enhances accuracy, as shown in Table R8. This approach is complementary to our key contributions. We sincerely thank the reviewer for highlighting this point. We cite the work [5] and update our result in our main tables. We name the method **Quamba-O**, indicating applying OCTAV on Quamba, in the tables.
>
> > The authors focus on the W8A8 case. yet Transformers have been quantized to 4-bit (e.g., using Quarot). Is it a fair comparison to use 8-bit transformers?
>
> We use the official implementation from QuaRot to profile TTFT for Llama-2-7B (W4A4KV4) on Nano 8G for reference purposes. The quantized W4A4KV4 Llama-2-7B is roughly comparable in model size to Quamba (8-bit Mamba), making it a reasonably equivalent basis for comparison.
> We report the TTFT in milliseconds (msec.) for all latencies on **Nano 8G** in Table R9. Empirically, we find that the 4-bit Llama-2-7B crashes with 8K input. We provide the theoretical memory cost for 4-bit Llama-2-7B in the updated Figure 1 (c).
>
> |Table R9                        | Bit-width | Model size | Latency (msec.)   | Acc. |
> |---------|-------|----------|----------|----|
> |                                      |                |                   | 1k &nbsp;&nbsp;&nbsp; &nbsp;&nbsp;&nbsp;/ 2k &nbsp;&nbsp;&nbsp; &nbsp;&nbsp;&nbsp;&nbsp;/ 4k &nbsp;&nbsp;&nbsp; &nbsp;&nbsp;&nbsp;&nbsp;/ 8k     |     |
> |Llama-2-7B  (QuaRot)   |W4A4KV4|3.5G          | 2430.6  / 4529.1 / 9212.8 / OOM    | 65.6%|
> |Quamba-2.8B-O            | W8A8      |2.7G          |1181.1  / 2354.9 / 4706.2 / 10888.9 | 62.5%|
>
> We compare our method against 4-bit Llama-2-7B on A5000 in Section C. As shown in Figure 7 in the section, we profile TTFT in milliseconds using 4K input tokens and profile TTLT in seconds with 2K input tokens and 2K generated tokens. We use QuaRot official implementation and profile latencies for Llama-2 on A5000 and Nano. **Our method offers the best trade-off between average accuracy and latency**, outperforming half-precision Mamba and Mamba-Had on both A5000 and Nano.
>
> To summarize our contributions and the focus of our work:
> - **Comparing against quantized Transformers is not our primary objective**, since SSMs do not use self-attention or the K-V cache. To provide a fair comparison and evaluate all current quantization techniques, we focus on techniques specifically designed or adapted for SSMs as our main baselines, such as Mamba-PTQ, Mamba-SmQ and Mamba-Had. The Transformers included in the figures and tables are provided for reference purposes only.
> - **The input context length for quantized Transformers remains constrained by the memory capacity of edge devices** (e.g., Orin Nano 4G/8G). While emerging techniques like 4-bit quantization (e.g., QuaRot) and K-V cache compression [8, 13] are improving Transformer efficiency, the memory complexity of Transformers remains linear with respect to the context length, as shown in Table R9. This makes the Transformer-based model inapplicable to long context summarization with limited computational resources.
> - **We demonstrate that QuaRot [13], the state-of-the-art W4A4 quantization method for transformer-based LLMs, is ineffective for SSMs at the 4-bit level**. As shown in Table R4, both Mamba-Had (QuaRot [13] re-implemented for Mamba) fail at the 4-bit level, reducing the average accuracy from 63.1% to 30.2%, respectively. This highlights the challenges SSMs present in model quantization under the low bit-width setting. We note that similar results were found recently for large transformer-based LLMs which were shown to have **best performance under memory constraints for a bit-width of 6-8, with worse results for a bit-width of 4 [14]**.
>
> | Table R1       | Bit-width | Avg. Acc. |
> |----------------|-----------|-----------|
> | Mamba-2.8B     | FP16      | 63.1%     |
> | Mamba-Had-2.8B | W4A4      | 30.2%     |

---

> > ### Author Response · Authors · 2024-11-21
> >
> > > Theorem 4.1 is very badly written. Lowercase b and epsilon were never defined. It seems like this was just added as decorative math.
> >
> > We present the theoretical proof for the **discrete 1D linear time-invariant (LTI) SSM**, as described on line 225, and therefore, the notation in the proof is **scalar-based**. We have updated our manuscript to further clarify this.
> >
> > The theorem is not merely decorative math. To the best of our knowledge, **no prior work has determined the theoretical error bound for the linear recurrence mechanism modeled by Equation 1**. To this end, we provide in Theorem 4.1 a theoretical foundation for quantizing SSMs to support and connect our work: The quantization error is bounded at time step $t$. This demonstrates that our study is grounded in **theoretical proof**, **empirically validated across multiple datasets**, and **practically accelerated on hardware platforms**.
> >
> >
> > > While I appreciate Theorem 4.1 which states that quantization error compounds over time-steps (correct me if I am wrong)
> >
> > **This is incorrect**; the theorem shows the error is **bounded**, not that it compounds over time-steps. The essence of Theorem 4.1 is to show that quantization error is bounded by $b \epsilon \frac{e^{t-T}}{e-1}$, where $t$ is the current step and $T$ is the input sequence length (i.e., the total steps). For any sequence of length $T$, the maximum per-token error occurs at the last token and is bounded by $b \epsilon \frac{1}{e-1}$, as $t$ equals $T$.
> >
> > Most importantly, we show that the global and per-token quantization error is **bounded**. For Mamba-style models, we conjecture that error **is not compounding** since a compressed model may not necessarily strictly deviate from the full precision model across time.

---

> > > ### Comment · Reviewer_evxv · 2024-11-21
> > > **Thank you + Follow-up**
> > >
> > > Good job on the responses. Overall this has addressed most of my concerns so I am happy to raise the score. But I still have lingering questions.
> > >
> > > In Theorem 4.1, epsilon is still not defined in the main text, it's only defined in the appendix. You shouldn't introduce an undefined term in a Theorem in the main text. Please define epsilon in the main text.
> > >
> > > For  this theorem, it is assumed that a(t,T) is a nicely behaving exponential function  that will always be smaller than 1. Thus it is not surprising that quantization noise does not explode when unrolling the recurrence back in time. I do not mind using this assumption for the theorem to work. However, can you please discuss how realistic is this assumption. In the real case, A would be a matrix containing model parameters so we cannot assert that it would be so nicely behaved. In addition, some discussion on how the provided scalar-based result roughly translates to the tensor-based form (which is more useful in a real setting) would be useful.

---

> ### Author Response · Authors · 2024-11-22
>
> We appreciate your recognition of our work and your positive feedback. We are also grateful for your rating, which leans towards accepting our submission. We provide responses to your follow-up questions below.
>
> > In Theorem 4.1, epsilon is still not defined in the main text
>
> Thanks for pointing this out. We have updated our manuscript and included the notation in the main body of the paper.
>
>
>
> > Can you please discuss how realistic is this assumption. In the real case, A would be a matrix containing model parameters so we cannot assert that it would be so nicely behaved.
>
>
> Mamba models utilize matrices A, B, C, and D, as defined in the original papers [15, 16]. Notably, we observe that HiPPO [16] exhibits similar characteristics to the exponential series, as illustrated in Figure 5 of [16].
>
> To address the reviewer's concern, we provide an empirical error analysis in Section A.2, where we instantiate the A and B matrices using the HiPPO-LegT and HiPPO-LegS formulations described in [16]. As shown in Figure 5, Section A.2, the errors remain bounded within the context of a discretized high-dimensional SSM.
>
> [15] Gu, Albert, and Tri Dao. "Mamba: Linear-time sequence modeling with selective state spaces." arXiv preprint arXiv:2312.00752 (2023).
>
> [16] Gu, Albert, et al. "Hippo: Recurrent memory with optimal polynomial projections." Advances in neural information processing systems 33 (2020): 1474-1487.

---

> > ### Comment · Reviewer_evxv · 2024-11-22
> > **Can the analysis on these matrices be tied to the statement of the theorem**
> >
> > It would be excellent if we could speak more to the generalizability of theorem 4.1 to the matrix case. What happens when unrolling the equation over time. Doe we get back-to-back-to----to-back GEMMs of the model matrix which then multiplies the error vector in a GEMV? In this case, do we get a condition on the spectral norm of the model matrices for the quantization noise to be bounded?

---

> ### Author Response · Authors · 2024-11-26
>
> Thank you for your follow-up questions, which prompted us to explore the matrix case of the proof and conduct a deeper analysis of the bound for quantization error.
>
> > What happens when unrolling the equation over time. In this case, do we get a condition on the spectral norm of the model matrices for the quantization noise to be bounded?
>
> We have followed your suggestion and included the results and proof for the matrix case. We have updated our proof for the matrix case in Section 4.1 in the main paper and Section G in Appendix.
>
> We consider a discrete linear time-invariant (LTI) state space model at time step $t$ defined as $h(t) = A(t) h(t-1) + B  x(t)$, where the state matrix $A(t) \in \mathbf{R}^{N \times N}$,  input matrix $B \in \mathbf{R}^{N \times P}$, implicit latent state $h(t) \in \mathbf{R}^{N \times 1}$, and input $x(t) \in \mathbf{R}^{P \times 1}$. $T$ represents the input sequence length and $1 \leq t \leq T$.
>
> We visualize the spectral norm of the cumulative product of $A$ matrices (i.e., $||A(t)A(t − 1)· · · ||_2$) from pre-trained Mamba-130m in Figure 10 in Section G.1. The plot shows that the **SSM models put more weight on recent history by performing a cumulative product of the $A$ matrix**. Due to this property of SSMs, we assume that the norm of the state matrix $A$ can be represented  by an **exponential function** in our proof such that $||A(t)||_2 \leq a \cdot e^{t-T}$, given $0<a<1$. Furthermore, we assume the input matrix $B$ is bounded by $||B||_2 \leq b$, where $b>0$.   In the proof, we assume the system input contains a quantization error $\delta_x(t)$, such that $\overline{x}(t) = x(t) +\delta_x(t)$, where $||\delta_x(t)||_2 \leq \epsilon$ and $\epsilon>0$. We utilize the spectral norm ($|| \cdot || = || \cdot ||_2$). The quantization error at time step $t$ is defined as $\Delta(t) = \overline{h}(t) - h(t)$.
>
> Following this observation and assumption, we show that the **time-step dependent bound becomes: $||\Delta(t)||_2 \leq \epsilon b (\frac{1}{1 - ae^{t-T}})$**. Therefore, the **global bound for a given sequence length $T$ (i.e., $t=T$) is $||\Delta(T)||_2 \leq  \frac{\epsilon b}{1 - a}$, given $0<a<1$**. As our Figure 10 shows, the spectral norm of the $A(t)$ matrix is upper bounded by 0.8  (i.e., $||A(t)||_2 \leq 0.8$) and the norm decays as the SSM performs the cumulative product of A matrices (i.e., $||A(t)A(t − 1)· · · ||_2$). Therefore, our quantization error is bounded with respect to the input sequence length $T$.
>
> Please see details in our updated manuscript for the matrix case in Section 4.1 in the main paper and Section G in Appendix.

---

> > ### Author Response · Authors · 2024-12-01
> >
> > Dear Reviewer evxv,
> >
> > We hope we have addressed all your concerns. We look forward to your response and any final discussions.

---

### Official Review · Reviewer_TXWC · 2024-11-02

**Soundness:** 2
**Presentation:** 2
**Contribution:** 3
**Rating:** 5
**Confidence:** 3

**Summary:**

This paper introduces a post-training quantization method tailored for Selective State Space Models (SSMs) to address challenges in model size reduction and computation efficiency, specifically in hardware-constrained environments. The authors propose an 8-bit static quantization approach that handles weight and activation quantization while mitigating the effects of outliers commonly found in SSM outputs.

Key contributions of the paper include:
1.  A method for input activation quantization that reduces the impact of extreme values by clipping based on the 99.999th percentile, improving quantization precision.
2. Hadamard transforms to map output activations into a space with minimized outliers, allowing for accurate quantization without excessive memory and computational cost.
3. Empirical analysis showing the effectiveness of this quantization approach in achieving a balance between model accuracy and reduced latency, making SSMs more feasible for edge and embedded applications.

The paper provides a comprehensive empirical evaluation, demonstrating how these techniques enable SSMs to perform efficiently on hardware with limited resources.

**Strengths:**

1. The paper presents a unique approach by applying quantization specifically to Selective SSMs, a topic with limited prior research in this area. The integration of percentile-based clipping and the Hadamard transform for managing outliers in SSMs represents an innovative combination of techniques. These methods are tailored specifically for the challenges of SSM quantization, marking a significant contribution to the field of model compression.
2. The paper demonstrates a thorough experimental setup, with well-defined quantization methodologies for activations. The authors support their approach with empirical evidence, clearly detailing the implementation of 8-bit static quantization and its impact on latency and accuracy. The careful construction of quantization techniques for hardware constraints is grounded in a rigorous understanding of SSM behavior, lending credibility to the study’s findings.
3. The proposed quantization approach has high potential significance, especially for applications involving SSMs in resource-limited environments. By addressing the specific challenges of quantizing SSMs, the paper extends the practical use of SSMs, making them feasible for deployment on low-power hardware without significant accuracy degradation.

**Weaknesses:**

The paper is innovative in its approach and thorough experimentation. However, there are several critical questions that I raised in the "Question" section, which I believe are essential for the clarity and robustness of the findings. I hope the authors can provide insights on these points, and I look forward to further discussion.

**Questions:**

1. As a researcher focused on large model quantization on edge devices, I have some concerns about the authors' claim regarding the lack of prominent outliers in transformers. Based on my own experiments and findings [1-2] from recent literature, larger language models (LLMs) actually tend to exhibit more significant outliers, especially in channel-wise activation. Can you clarify your claim about outliers in Transformers and provide evidence or citations supporting your view?

2. Studies [3-4] on transformer quantization have shown that quantizing outliers can substantially impact model accuracy. This observation aligns with the authors' ablation study results. However, I am unclear on why certain techniques in this paper appear to alleviate the impact of outlier quantization. Additionally, shouldn’t each optimization step incrementally reduce accuracy? Surprisingly, the authors' ablation study shows an improvement in accuracy when all three techniques are applied together. Could you please provide a more detailed explanation of why your techniques, when combined, lead to improved accuracy rather than incremental reductions?

3. In the introduction, the authors mention that this quantization method is applicable to both edge devices and cloud computing. However, the experiments only feature the high-compute edge device Orin Nano 8G (and the A5000, which would generally not be considered an edge device). There is no testing on cloud computing hardware, which would have been helpful.

4. For the results in Table 1: Profiling Latency, I find some inconsistencies. As token length increases, both computation and memory should scale linearly, so I would expect a relatively stable speedup ratio. If anything, given the execution overhead of the nonlinear kernel within SSM, there might be a slight increase in latency with token length. However, the authors' data show an unusual spike in performance at $L=512$ and $L=1$. Could you please provide a more detailed explanation of the observed performance spikes, possibly including additional data points or experiments that could help clarify the relationship between token length and speedup ratio?

5. Lastly, could the authors clarify how many hardware measurements were conducted to obtain the reported results? Given the inherent variability in hardware performance, multiple measurements are typically necessary to ensure stability and reliability. It would be helpful to understand the stability of the hardware results presented. Could you please provide details on the number of runs performed for each experiment, any measures taken to account for variability (e.g., averaging results, reporting standard deviations), and how you ensured the stability and reproducibility of your hardware measurements?

[1] Shen, Xuan, et al. "Agile-Quant: Activation-Guided Quantization for Faster Inference of LLMs on the Edge." Proceedings of the AAAI Conference on Artificial Intelligence. Vol. 38. No. 17. 2024.
[2] Xiao, Guangxuan, et al. "Smoothquant: Accurate and efficient post-training quantization for large language models." International Conference on Machine Learning. PMLR, 2023.
[3] Lin, Yang, et al. "Fq-vit: Post-training quantization for fully quantized vision transformer." arXiv preprint arXiv:2111.13824 (2021).
[4] Dong, Peiyan, et al. "Packqvit: Faster sub-8-bit vision transformers via full and packed quantization on the mobile." Advances in Neural Information Processing Systems 36 (2024).

---

> ### Author Response · Authors · 2024-11-21
>
> We deeply appreciate your detailed review of our submission and provide responses to your concerns and questions below.
>
>
> > the authors' claim regarding the lack of prominent outliers in transformers.
>
> **We do not make this claim in our study**. Instead, we show that the **outlier patterns in SSMs and Transformers differ**. In line 22, we note “massive outliers in output activations which are not present **in the output token-mixing** of self-attention modules”. This observation aligns with the findings of [1] and [2], which indicate that channel-wise outliers in Transformers primarily occur in the **feedforward layer**, leading to significant accuracy loss when quantizing activation outputs from linear layers. In contrast, quantizing self-attention layers (responsible for token mixing) in Transformers does not affect accuracy, as shown in the $y$ column in Figure 10 (b) and (c). For Mamba, however, outliers appear in the output of the SSM (i.e., linear recurrence mechanism for token-mixing).
>
> Our study finds that the output of the **SSM output $y$ is highly sensitive to the quantization errors of input $x$ due to the causal relationship** modeled in Equation 1.
> We tackle this hard problem and employ clipping and Hadamard transform to address the issues in SSMs, **not because they are easy, but because they are effective in terms of performance and latency** with suitable non-trivial changes. We validate Quamba’s practicality with **1.7× and 1.2× speedups on edge and cloud GPUs**, respectively, while maintaining accuracy across multiple tasks. Furthermore, our method provides average accuracy and latency that are on the **Pareto front when compared to Transformer baselines (as shown in  in Appendix C, Figure 7), highlighting its uniqueness and novelty in SSM quantization**. We report seconds (sec.) for all time-to-last-token (TTLT) latencies and the average accuracy among six zero-shot tasks in Table R2.
>
> | Table R2             | Bit-width  |  Latency (sec.)   |Avg. Acc.  |
> |-----------------------|:--:|-----------------------|-----------|
> |                            |  | 1k &nbsp; / 2k &nbsp; / 4k &nbsp; &nbsp; /  8k    |        |
> | Mamba-2.8B      | FP16 | 5.1 / 10.3 / 20.7 / 41.5  |  63.1%    |
> | Mamba-Had       | W8A8 | 5.5 / 10.8 / 21.9 / 43.8  |  62.4%    |
> | Quamba-2.8B     | W8A8  | **4.1** / **8.4** &nbsp; / **16.9** / **33.8**  |  62.2%    |
> | Quamba-2.8B-O  | W8A8  | **4.1** / **8.4** &nbsp; / **16.9** / **33.8**  |  **62.5%**    |
>
> In response to the reviewers’ suggestions, we applied a clipping optimization algorithm [5] to Quamba, naming the enhanced version Quamba-O. This modification improves our average accuracy and outperforms Mamba-Had in both accuracy and latency. We highlight our innovations in proposing an efficient quantization framework for SSMs that achieves Pareto optimality on both edge and cloud GPUs. Optimizing clipping range is complementary to our key contributions. We sincerely thank the reviewer for highlighting this point. We cite the work [5] and update the result in our tables.
>
>
> > unclear on why certain techniques in this paper appear to alleviate the impact of outlier quantization.
>
> **We employ clipping to improve the input’s quantization precision thereby reducing the deviation at the SSM output**. We find that SSM output $y$ is sensitive to the quantization errors of input $x$ (i.e., the quantization errors of x leading to large deviations in output y) due to the causal relationship modeled in Equation 1, as shown in Figure 3 (a).
>
> **We quantize y in a smoother space by applying Hadamard matrices to manage the outliers in the $y$ tensor**, thereby improving the quantization precision, as shown in Figure 3 (b).
>
> Apart from the large outliers in the $y$ tensor output from the SSM, our study shows that the additional challenge posed in SSMs is maintaining quantization precision for the $x$ tensor input to the linear recurrence.
>
> > The ablation study shows an improvement in accuracy when all three techniques are applied together, even though each technique incrementally reduces accuracy
>
> We begin with naive W8A8 quantization and **progressively apply our techniques** to enhance average accuracy. For the 2.8B model, naive W8A8 achieves only 45.9% average accuracy. Adding clipping improves accuracy to 48.5%, and incorporating the Hadamard transform increases it to 56.4%. Finally, applying both techniques yields our Quamba accuracy of 62.2%. This demonstrates that 8-bit quantization for SSMs is non-trivial, and our method effectively closes the performance gap between the 8-bit model and the half-precision model. Table R5 is extracted from Table 5 in our paper.
>
> | Table R5        |  FP16 | Naive W8A8 | + Clip x |+ Had. y | + Both |
> |-------------------|----------|:----------------:|----------|---------|--------|
> | Mamba 2.8B | 63.0% |  45.9%     |  48.5%   | 56.4%   | 62.2%  |

---

> ### Author Response · Authors · 2024-11-21
>
> > There is no testing on cloud computing hardware, which would have been helpful.
>
> For the scope of this work, we believe **A5000 performance is a satisfactory proxy for “cloud” performance**. We test Quamba on A5000, a common AI workload GPU with 24GB memory, in our lab’s AI workload cluster with computing power comparable to its data center counterparts (e.g., A10). We have updated our manuscript to further clarify it in line 353-355. We will open-source our implementation for users to test on various platforms.
>
> > The authors' data show an unusual spike in performance at L=512 and L=1.
>
> This may result from our tensor core or thread configurations particularly fitting the L=512 case.  Regarding L=1, we use the GeMV kernel in the generation stage (L=1) as the PyTorch backend does. Since the generation stage (i.e., decoding) is memory-bound, quantizing weights to 8-bit halves memory reads, thereby enhancing computational efficiency. The PyTorch framework employs different matrix multiplication kernels with various tensor core shapes and thread warp size configurations in the backend, depending on input characteristics. We did not adjust the tensor core or thread configurations for each case, nor did we disable the PyTorch cuDNN auto-tuner, which highlights the robust acceleration achieved by our method. We have updated our manuscript to further clarify it in line 349. We will open-source our implementation for other researchers to reproduce our experiments.
>
>
>
> > Could you please provide details on the number of runs performed for each experiment
>
> We report the Mean ± Std in Table 6 on A5000.
>
> |Table R6   |Bit-width|Generate(L=1)|   L=512   |   L=1024  | L=2048 |
> |-----------|:---------:|:-------------:|:-----------:|:-----------:|:--------:|
> |Mamba-2.8B |  FP16   | 9.86±0.038|61.19±2.79|102.29±1.42|184.16±1.89|
> |Quamba-2.8B|  W8A8   | 8.12±0.011|48.24±1.53 |84.84±0.46|165.13±0.81|
>
> For our setup, we perform 10 warm up iterations and report the average inference time over the next 100 runs, as we state in line 356. We follow the Pytorch profiler document to implement our timer. Specifically for the generation stage (L=1), we compile the inference CUDA graph for the model to eliminate additional CPU-GPU interactions. We ensured that the elapsed time was recorded only after all CUDA kernels were synchronized and completed. We will open-source our implementation for other researchers to reproduce our experiments.

---

> > ### Author Response · Authors · 2024-11-27
> > **Friendly Deadline Reminder: Discussion Period Ends December 2 (AoE)**
> >
> > Dear Reviewer TXWC,
> >
> > We are also looking forward to further discussing your questions and concerns. As a friendly reminder, the discussion period ends on December 2nd (AoE).

---

> > > ### Author Response · Authors · 2024-12-01
> > >
> > > Dear Reviewer TXWC,
> > >
> > > We hope we have addressed all your concerns. We look forward to your response and any final discussions.

---

### Official Review · Reviewer_PfeA · 2024-11-02

**Soundness:** 3
**Presentation:** 3
**Contribution:** 4
**Rating:** 8
**Confidence:** 3

**Summary:**

This work propose a PTQ solution for SSM model. Quamba notices the SSMs have highly sensitive feature maps within the selective scan mechanism (i.e., linear recurrence) and massive outliers in the output activations, and utilises two methods of outlier suppression - ①suppresses the maximum values of the input activations ②supply hadamard transform of the output activations, to solve the problem. Experiments show that show that this method can achieve 8-bit PTQ quantisation of SSM almost losslessly.

**Strengths:**

- The approach is simple and effective, introducing little additional overhead
- The experiment is solid: The author achieved real quantization - by handwriting fused operators and utilizing low-bit computational libraries (e.g. cutlass) to implement Quamba.

**Weaknesses:**

- The proposed method is effective, however, these methods are also widely used in Transformer, and authors may need to make their connection to SSM more explicit in their writing.

**Questions:**

- In line 228, you mentioned the **Theoretical error bound for SSM quantization**, and give a proof in appendix. However, this discussion seems to be isolated, so I do not particularly understand the connection between this discussion and the core approach of this paper.
- In Table 1 and Table 2, you compared QuaRot and Quamba in W8A8 format. However, QuaRot mainly focus on **W4A4** solution for quantization. I would like to know if this method can be extended to W4A8 and W4A4 scenarios? If it can, how is his performance compared to QuaRot?

---

> ### Author Response · Authors · 2024-11-21
>
> We sincerely appreciate your thorough review of our submission and address the concerns and questions below.
>
> > These methods are also widely used for transformers
>
> While we include existing methods introduced for transformer quantization as our baselines for naive Mamba quantization, we argue that the methods widely used for Transformers **do not transfer well** to SSMs. Table R1, 7 and 8 demonstrate this. For example, Mamba-Had **fails** to achieve meaningful perplexity and reduces average accuracy from 63.1% to 30.2% on the six zero-shot tasks in its original setting (W4A4). This shows quantizing SSM is specifically challenging and not yet satisfactorily addressed. Table R1 below is extracted from Table 8 in our paper.
>
> | Table R1        | Bit-width | Avg. Acc. |
> |-----------------|-----------|-----------|
> | Mamba-2.8B      | FP16      | 63.1%     |
> | Mamba-Had-2.8B  | W4A4      | 30.2%     |
>
> We tackle the poor generalizability of transformer quantization methods to SSMs and employ clipping and Hadamard transform to address these issues for SSMs. Taken together, these two modifications are **effective** in maintaining **performance** and **reducing latency** with suitable non-trivial changes. The transformer quantization approach described in [13] re-implemented for SSMs and designated as **Mamba-Had (W8A8) does not offer latency benefits** over the FP16 model on A5000, as shown in Table R2 and the updated Figure 1 (b). In contrast, **Quamba offers 1.7× and 1.2× speedups** on edge and cloud GPUs, significantly outperforming Mamba-Had. Furthermore, our method achieves a Pareto-optimal balance between accuracy and latency compared to Transformer baselines, **highlighting its uniqueness and novelty in SSM quantization**. We report seconds (s) for all time-to-last-token (TTLT) latencies on A5000 and the average accuracy among six zero-shot tasks in Table R2.
>
> | Table R2             | Bit-width  |  Latency (sec.)   |Avg. Acc.  |
> |-----------------------|:--:|-----------------------|-----------|
> |                            |  | 1k &nbsp; / 2k &nbsp; / 4k &nbsp; &nbsp; /  8k    |        |
> | Mamba-2.8B      | FP16 | 5.1 / 10.3 / 20.7 / 41.5  |  63.1%    |
> | Mamba-Had       | W8A8 | 5.5 / 10.8 / 21.9 / 43.8  |  62.4%    |
> | Quamba-2.8B     | W8A8  | **4.1** / **8.4** &nbsp; / **16.9** / **33.8**  |  62.2%    |
> | Quamba-2.8B-O  | W8A8  | **4.1** / **8.4** &nbsp; / **16.9** / **33.8**  |  **62.5%**    |
>
> In response to the reviewers’ suggestions, we applied a clipping optimization algorithm [5] to Quamba, designating the enhanced version as **Quamba-O**. This modification improves our average accuracy and outperforms Mamba-Had in both accuracy and latency. We highlight our innovations in proposing an **efficient quantization framework for SSMs** that achieves Pareto optimality on both edge and cloud GPUs. Optimizing clipping range is complementary to our key contributions. We sincerely thank the reviewer for highlighting this point. We cite the work [5] and update the result in our tables.
>
>
>
> > Quamba’s connection with SSMs should be more explicit.
>
> **Our innovation starts with a key observation from SSM in Figure 2, which establishes a connection to SSM**: Quantizing SSM input leads to large deviations in the output due to the linear recurrence and SSM output has large outliers. Input sensitivity is unique to SSMs since self-attention exhibits more robustness to input quantization,  as shown in Figure 2. We present an in-depth analysis of the causal relationship between input and output activations in SSMs and Transformers in Figure 2 and Appendix Section I.
>
> **Our technique addresses this specific challenge in SSMs: input sensitivity**. In Quamba, we enhance quantization precision for the x tensor and reduce the deviation in the SSM output. To deal with the outliers in SSM output, we employ Hadamard transform to eliminate the outliers. Most critically, our method is hardware friendly and offers 1.7× and 1.2× speedup on edge and cloud GPUs, both of which are on the Pareto front in average accuracy and latency when compared with Mamba-Had and its Transformer counterparts, as shown in  in Appendix C, Figure 7.
>
> > Connection between the theoretical error bound for SSM quantization and the core approach of this paper.
>
> To the best of our knowledge, **no prior work has studied the theoretical error bound of the linear recurrence mechanism modeled by Equation 1**. To this end, we provide a theoretical foundation for quantizing SSMs to support and connect our work: **Theorem 4.1 shows that the quantization error is bounded for any input sequence length T**. This demonstrates that our study is grounded theoretically, empirically validated across multiple datasets, and practically accelerated on both cloud and edge hardware platforms.

---

> ### Author Response · Authors · 2024-11-21
>
> > In Table 1 and Table 2, you compared QuaRot and Quamba in W8A8 format. However, QuaRot mainly focuses on W4A4 solution for quantization.
>
> **We demonstrate that QuaRot is ineffective for SSMs under the 4-bit setting**. As shown in Table R1 and Tables 7 and 8 (in the main paper), Mamba-Had implemented for W4A4 reduces the original Mamba average accuracy from 63.1% to 30.2%. Consequently, we do not consider W4A4 configuration in the remainder of the paper and only use the approach presented in [13] for the W8A8 case. We note that similar results were found recently for large transformer-based LLMs which were shown to have **best performance under memory constraints for a bit-width of 6-8, with worse results for a bit-width of 4 [14]**. Table R1 is extracted from Table 8 in our paper.
>
> | Table R1       | Bit-width | Avg. Acc. |
> |----------------|-----------|-----------|
> | Mamba-2.8B     | FP16      | 63.1%     |
> | Mamba-Had-2.8B | W4A4      | 30.2%     |
>
> **To alleviate the reviewer’s concern and provide an additional comparison**, we include SmoothQuant [12], the state-of-the-art W8A8 quantization method, in Table 1 and 2 to compare  latency and accuracy. Applying SmoothQuant to SSM linear layers does not introduce additional overhead but results in a significant degradation in average accuracy. In contrast, Quamba achieves **the best trade-off between average accuracy and latency** on Orin Nano 8G. We summarize these results in Table R3 and Figure 1 (a). Table R3 is extracted from Table 1 in our paper.
>
> | Table R3        | Bit-width   | Speedup (×) | Avg. Acc.  |
> |------------------|--|-------------|------------|
> | Mamba-2.8B | FP16   | 1.0  ×      | 63.1%      |
> | Mamba-SmQ-2.8B | W8A8   | 1.83 ×      | 57.3%      |
> | Mamba-Had-2.8B  | W8A8    | 1.52 ×      | 62.4%      |
> | Quamba-2.8B       | W8A8  | **1.72 ×**      | 62.2%      |
> | Quamba-2.8B-O     | W8A8  | **1.72 ×**      | **62.5%**      |
>
>
> > I would like to know if this method can be extended to W4A8 and W4A4 scenarios? If it can, how is his performance compared to QuaRot?
>
> We have evaluated both Quamba and Mamba-Had under the W4A4 setting.  However, as shown in Table R4, **both** Mamba-Had (QuaRot [13] re-implemented for Mamba) and Quamba **fail at the 4-bit level**, reducing the average accuracy from 63.1% to 30.2% and 29.3%, respectively. **This highlights the challenges SSMs present in model quantization under the low bit-width setting**. We note that similar results were found recently for large transformer-based LLMs which were shown to have **best performance under memory constraints for a bit-width of 6-8, with worse results for a bit-width of 4 [14]**.
>
> | Table R4        | Bit-width | Avg. Acc. |
> |-----------------|-----------|-----------|
> | Mamba-2.8B      | FP16      | 63.1%     |
> | Mamba-Had-2.8B  | W4A4      | 30.2%     |
> | Quamba-2.8B     | W4A4      | 29.3%     |
>
> As a result, **our work focuses on SSM quantization under the W8A8 setting**. In this case, **Quamba clearly outperforms Mamba-Had by achieving 1.7× and 1.2× speedups** on edge and cloud GPUs. In contrast, Mamba-Had (W8A8) fails to provide latency benefits over the FP16 model on the A5000, as shown in Table R2 and the updated Figure 1(b). We report seconds (s) for all time-to-last-token (TTLT) latencies on A5000 and the average accuracy among six zero-shot tasks in Table R2.
>
> | Table R2             | Bit-width  |  Latency (sec.)   |Avg. Acc.  |
> |-----------------------|:--:|-----------------------|-----------|
> |                            |  | 1k &nbsp; / 2k &nbsp; / 4k &nbsp; &nbsp; /  8k    |        |
> | Mamba-2.8B      | FP16 | 5.1 / 10.3 / 20.7 / 41.5  |  63.1%    |
> | Mamba-Had       | W8A8 | 5.5 / 10.8 / 21.9 / 43.8  |  62.4%    |
> | Quamba-2.8B     | W8A8  | **4.1** / **8.4** &nbsp; / **16.9** / **33.8**  |  62.2%    |
> | Quamba-2.8B-O  | W8A8  | **4.1** / **8.4** &nbsp; / **16.9** / **33.8**  |  **62.5%**    |

---

> > ### Comment · Reviewer_PfeA · 2024-11-24
> >
> > Thanks for the detailed rebuttal. i see from it that the author's implementation is very SOLID, and I think similar work about light-weight inference is more meaningful in the implementation of an indestructible system than algorithmic novelty and theoretical rigour. I have therefore decided to raise the score, although I would like to give the authors two pieces of advice:
> > 1. The highlight of this work is in the **system implementation** and **evaluation** rather than in the **theoretical proof**. Therefore, I think the theoretical proofs that have little to do with the system implementation can be further abbreviated or even put in full in the appendix.
> > 2. Accordingly, some parts of the appendix related to system implementation (such as on edge devices) can be brought forward, because as far as I know, many papers associated with lightweight machine learning only simulate the algorithms, but rarely deploy them on real hardware.

---

> > > ### Author Response · Authors · 2024-11-26
> > >
> > > We appreciate your careful review and encouraging rating, indicating support for our paper.
> > >
> > > > Some parts of the appendix related to system implementation (such as on edge devices) can be brought forward. The theoretical proofs that have little to do with the system implementation can be further abbreviated or even put in full in the appendix.
> > >
> > > We have followed your suggestion and included an abbreviated Pareto front analysis in Section 5.2 of the main paper. The full Pareto front analysis, along with other system implementation details, is in the Appendix. We have condensed our proof and highlighted the conclusions of our theoretical analysis in the main paper.

---

### Author Response · Authors · 2024-11-21

We would like to thank all reviewers for reading our paper and for their insightful feedback. We are excited to see the interest in quantizing SSMs/Mamba style LLMs as evidenced by other concurrent work on the topic. We note our work provides **the only** comprehensive analysis of quantization on Mamba models not only on **accuracy** and resulting **lower model size**, but also on the significantly **lower latency** achieved on both **high-end** and **edge** GPU platforms for both **pre-filling** and **generation** stages.


## Key Modifications
We would like to note that QuaRot-SSM in our paper is **not** QuaRot [13] but rather the Mamba architecture with the Hadamard transform previously introduced for transformer LLMs in [13]. This modified Mamba architecture constitutes one of our baselines and is shown in Figure 7. However, we find it particularly **inefficient in latency** as shown in Table R8. To avoid confusion, **we replaced QuaRot-SSM with Mamba-Had** throughout the paper and in this rebuttal. Similarly, we replace SmQ-SSM to Mamba-SmQ, which is the Mamba architecture modified with the approach introduced in SmoothQuant [12]. This is also one of our baselines.

In response to reviewers’ suggestions, we also applied a clipping optimization algorithm [5] to our proposed Quamba approach, with the new  version identified as **Quamba-O**. This modification improves our average accuracy and just like Quamba, outperforms Mamba-Had in both accuracy and latency.

We report seconds (s) for all time-to-last-token (TTLT) latencies on A5000 and the average accuracy among six zero-shot tasks in Table R8. As it can be seen, Quamba and Quamba-O enjoy **1.24-1.35 speedup** when compared to original Mamba or Mamba-Had which implements the quantization approach from [13]. All quantized models in Table R8 below use a W8A8 quantization scheme.

|     Table R8  | 130m  | 370m  | 1.4B  | 2.8B  | 1K latency |
|--------------|-------|-------|-------|-------|-----------|
| Mamba        | 44.7% | 51.7% | 59.7% | 63.1% |    5.1    |
| Mamba-Had    | 42.6% | 50.0% | 58.1% | 62.4% |    5.5    |
| Quamba       | **43.5%** | 49.0% | **58.4%** | 62.2% |    4.1    |
| Quamba-O     | 42.9% | **50.4%** | **58.4%** | **62.5%** |    4.1    |

Our innovations stem from proposing an **efficient quantization framework for SSMs** which puts our approach on the Pareto front of latency and accuracy for both edge and cloud GPUs. This is summarized in Appendix C, Figure 7.
Optimizing clipping range is complementary to our key contributions. We sincerely thank the reviewer for highlighting this point. We cite the work [5] and update the results in our tables.

## Other Modifications
We outline the following main changes in the submission PDF (highlighted in blue in the manuscript) and detailed in this rebuttal:

- Rename QuaRot-SSM to Mamba-Had
- Rename SmQ-SSM to Mamba-SmQ
- Add Section 5.2, Section C in the Appendix, and Figure 7 to show Pareto-front (reviewer evxv, PfeA)
- Figure 1 (b): Add Mamba-Had (reviewer 8GLX)
- Figure 1 (c): Add Llama-2-7B W4A4KV4, the original QuaRot approach targeted for transformer-based LLMs [13] (reviewer evxv)
- Figure1, Table3 and Table 6: Add Quamba-O (reviewer evxv)
- Figure 2: Correct the y-label (reviewer evxv)
- Section 4.1: Clarify our proof (reviewer evxv and PfeA)
- Add Section H: Extend our proof to the matrix case (reviewer evxv)
- Section 5: Clarify latency profiling (reviewer TXWC)

---

> ### Author Response · Authors · 2024-11-21
>
> ## List references used in the rebuttal
> [1] Shen, Xuan, et al. "Agile-Quant: Activation-Guided Quantization for Faster Inference of LLMs on the Edge." Proceedings of the AAAI Conference on Artificial Intelligence. Vol. 38. No. 17. 2024.
>
> [2] Xiao, Guangxuan, et al. "Smoothquant: Accurate and efficient post-training quantization for large language models." International Conference on Machine Learning. PMLR, 2023.
>
> [3] Lin, Yang, et al. "Fq-vit: Post-training quantization for fully quantized vision transformer." arXiv preprint arXiv:2111.13824 (2021).
>
> [4] Dong, Peiyan, et al. "Packqvit: Faster sub-8-bit vision transformers via full and packed quantization on the mobile." Advances in Neural Information Processing Systems 36 (2024).
>
> [5] Sakr, Charbel, et al. "Optimal clipping and magnitude-aware differentiation for improved quantization-aware training." International Conference on Machine Learning. PMLR, 2022.
>
> [6] Li, Rundong, et al. "Fully quantized network for object detection." Proceedings of the IEEE/CVF conference on computer vision and pattern recognition. 2019.
>
> [7] Lin, Yang, et al. "Fq-vit: Post-training quantization for fully quantized vision transformer." arXiv preprint arXiv:2111.13824 (2021).
>
> [8] Tang, Hanlin, et al. "Razorattention: Efficient kv cache compression through retrieval heads." arXiv preprint arXiv:2407.15891 (2024).
>
> [9] Lieber, Opher, et al. "Jamba: A hybrid transformer-mamba language model." arXiv preprint arXiv:2403.19887 (2024).
>
> [10] https://huggingface.co/ai21labs/Jamba-v0.1
>
> [11] Zhao, Han, et al. "Cobra: Extending mamba to multi-modal large language model for efficient inference." arXiv preprint arXiv:2403.14520 (2024).
>
> [12] Xiao, Guangxuan, et al. "Smoothquant: Accurate and efficient post-training quantization for large language models." International Conference on Machine Learning. PMLR, 2023.
>
> [13] Ashkboos, Saleh, et al. "Quarot: Outlier-free 4-bit inference in rotated llms." arXiv preprint arXiv:2404.00456 (2024).
>
> [14] Kumar, Tanishq, et al. "Scaling Laws for Precision." arXiv preprint arXiv:2411.04330 (2024).
>
> [15] Gu, Albert, and Tri Dao. "Mamba: Linear-time sequence modeling with selective state spaces." arXiv preprint arXiv:2312.00752 (2023).
>
> [16] Gu, Albert, et al. "Hippo: Recurrent memory with optimal polynomial projections." Advances in neural information processing systems 33 (2020): 1474-1487.

---

> > ### Author Response · Authors · 2024-12-01
> >
> > We sincerely thank all reviewers for their constructive feedback. Your insights have been invaluable in helping us refine our work and enhance the manuscript. We have incorporated your suggestions and comments, along with additional experiments and tables, into our updated paper. These changes are highlighted in blue.
> >
> > Thank you once again for your time and effort in reviewing our submission.

---

### Meta-Review · Area_Chair_BFMC · 2024-12-20

**Metareview:**

This paper presents an imporant contribution to SSM quantization, enabling faster inference with minimal accuracy loss. The authors carefully addressed initial reviewer concerns about novelty and theoretical clarity through additional experiments and improved explanations. Their framework, Quamba, shows strong latency reductions on real hardware without sacrificing performance. Overall, reviewers moved towards consensus after the rebuttal, recognizing the paper’s practical value and rigorous evaluation. While some minor points remain, they are not dealbreakers. The combined impression is that this work is both timely and impactful, making it deserving of acceptance.

**Additional Comments On Reviewer Discussion:**

Reviewers raised several key concerns: (1) unclear novelty compared to transformer quantization methods, (2) lack of advantage over QuaRot-SSM baseline, (3) methodology questions about latency measurements and MSE calculations, and (4) theoretical foundations needing clarification. The authors addressed these by: renaming QuaRot-SSM to Mamba-Had for clarity, adding Quamba-O variant with optimized clipping, providing detailed latency measurements, expanding theoretical proofs for matrix cases, and demonstrating clear speed advantages (1.7x on edge, 1.2x on cloud) over baselines. Reviewers were generally satisfied with the responses, with three raising their scores positively, citing improved clarity and solid implementation details.

---

### Decision · Program_Chairs · 2025-01-22

Accept (Poster)